# Sex-Specific Effect of a High-Energy Diet on Body Composition, Gut Microbiota, and Inflammatory Markers in Rats

**DOI:** 10.3390/nu17071147

**Published:** 2025-03-26

**Authors:** Dulce M. Minaya, Adam Hoss, Ayushi Bhagat, Tai L. Guo, Krzysztof Czaja

**Affiliations:** 1Department of Nutritional Sciences, College of Family and Consumer Sciences, University of Georgia, Athens, GA 30605, USA; dulce.minayacaba@uga.edu; 2Department of Biomedical Sciences, College of Veterinary Medicine, University of Georgia, Athens, GA 30602, USA; adam.hoss@uga.edu (A.H.); tlguo1@uga.edu (T.L.G.); 3Department of Pharmaceutical and Biomedical Sciences, College of Pharmacy, University of Georgia, Athens, GA 30602, USA; ayushi.bhagat@uga.edu

**Keywords:** obesity, high-energy-density diet, inflammatory cytokines/chemokines, gut dysbiosis, fecal lipid excretion

## Abstract

**Background/Objectives**: A high-energy-density (HED) diet promotes body weight gain, fat accumulation, and gut dysbiosis, contributing to obesity. The aim of this study was to characterize the initial response to HED diet consumption, as well as identify any sex differences in body composition, systemic inflammation, gut microbiome, and fecal fat excretion in rats. **Methods**: Male and female Sprague-Dawley rats were fed a low-energy-density (LED) diet for 10 days and were then switched to an HED diet for four weeks. Food intake, body weight, and body composition were measured routinely. Serum samples were collected to measure inflammatory cytokines/chemokines. Fecal samples were collected for microbiome analysis and lipid content. **Results**: After the HED diet, all rats gained body weight and fat mass, with males exhibiting increased susceptibility to weight gain. Males displayed either a diet-induced obesity phenotype (DIO-P) or a diet-resistant (DR) phenotype, as characterized by their differential body weight gain. Males showed elevated TGF-β levels, while females exhibited increases in Interferon gamma-inducible protein 10 (IP-10), regulated on activation, normal T cell expressed and secreted (RANTES) protein, and basic fibroblast growth factor (FGFb). Changes in gut microbiota composition revealed a reduction in beneficial species, like *Bacteroides uniformis* and *Parabacteroides distasonis*, and an increase in species such as *Akkermansia muciniphila*. Sex differences in fat metabolism were shown in the greater fecal fat excretion observed in males. **Conclusions**: Our study demonstrates that short-term consumption of a high-energy diet elicits notable sex-specific differences in body weight, body composition, inflammatory markers, gut microbiota, and fat excretion in Sprague-Dawley rats. While we recognize that this study has a small sample size and a short-term intervention, our findings highlight the critical role of sex as a biological variable in diet-induced obesity research.

## 1. Introduction

Obesity is a chronic, multifactorial disease characterized by excessive fat accumulation and low-grade inflammation, leading to health impairments [1]. It has been associated with several comorbidities, including hypertension, hyperglycemia, dyslipidemia, cardiovascular disease, type 2 diabetes, and cancer [1,2]. It is widely recognized that long-term consumption of a high-energy-density (HED) diet, high in fat and sugar, increases caloric intake, body weight, and fat mass accumulation [3,4]. It has also been shown to trigger microbiota dysbiosis [5], low-grade systemic inflammation [6], disruptions of the vagal gut–brain axis [7], and the mesolimbic reward pathway [8]. In addition, Sprague-Dawley rats typically exhibit a diet-induced obesity-prone (DIO-P) or diet-induced obesity-resistant (DR) phenotype when exposed to an HED diet [9,10]. Rats exhibiting the DIO-P phenotype have significantly increased body weights, caloric intakes, and fat mass compared to control or DR rats [11,12,13]. The emergence of these two phenotypes typically happens within five weeks of HED diet consumption [13,14].

HED diet consumption in animals has been shown to lead to dysbiosis in the gut, characterized mainly by an overall decrease in bacterial diversity and an increase in the ratio of *Firmicutes* to *Bacteroidetes* [15,16]. The *Firmicutes/Bacteroidetes* ratio has been proposed as a potential biomarker for obesity because studies have shown that the gut microbiota composition of obese animals and humans exhibits an increased abundance of firmicutes and decreased abundance of Bacteroidetes compared to their lean counterparts [17,18]. In addition, in a study comparing the gut microbiome of male and female rats exposed to an HED diet for three weeks, males had a greater increase in the *Firmicutes* to *Bacteroidetes* ratio than female rats [19]. While some bacterial populations can elicit detrimental responses in the host organism, others have been shown to be beneficial. The *Lactobacillus* genera, belonging to the *Firmicutes* phylum, and *Bifidobacterium*, belonging to the *Actinobacteria* phylum, are commonly used probiotics due to their ability to reduce inflammation, food intake, and body weight gain [20]. These observations serve as a possible link between gut dysbiosis and systemic inflammation in obesity.

The amount of fat excreted in feces after an HED diet can be used to determine the amount of dietary fat absorbed in the GI tract [21]. The small intestine is responsible for dietary fat absorption; however, overconsumption of dietary fat has been directly associated with increased intestinal permeability and disruptions in the proper functioning of the intestinal barrier, which can lead to intestinal disease and nutrient malabsorption [22]. However, there is limited and conflicting information available regarding the effects of the chronic overconsumption of dietary fat on the amount of fat excreted in stools. When fed a Western-type HED diet, female mice had a slightly greater fecal excretion of acylglycerol and cholesterol than male mice but had slightly less excretion of free fatty acids than males [23]. However, in another study of male and female mice fed a 20% fat HED diet, male mice had a greater fecal excretion of total triglycerides and total cholesterol than females [24]. In addition, in mice, Wang et al. showed that a long-term HF-diet intervention led to distinct metabolic adaptations in males compared to females [25].

A significant proportion of research in the field has predominantly focused on male subjects, leading to a considerable gap in our understanding of sex-specific responses and mechanisms. Studies comparing diet-induced obesity between male and female rats and mice are very few and often yield conflicting results. It has been reported that female rats are more susceptible to metabolic challenges [26,27,28,29], while research in mice generally indicates that females are either equally or less prone to dietary obesity [30,31,32]. Male rats and mice consume more food than their female counterparts, but the mechanisms or eating patterns underlying this sex difference may vary between species. Male rats typically consume larger meals, whereas male mice eat meals of the same size as female mice but more frequently [33,34,35]. In addition, major differences in study designs, with a variety of diet types, onset of intervention, and duration of diet, make it difficult to tease apart what variables cause the observed sex differences and which model best corresponds to sex differences seen in human obesity. To our knowledge, only one prior study has systematically, under the same methodological conditions, evaluated the effects of high-fat feeding on both rats and mice, including females and males of each species [36]. In this study, we sought to evaluate the effect of short-term HED diet consumption on fat mass accumulation, microbiota composition, systemic inflammation, and fecal fat excretion in male and female rats. Our goal was to characterize the initial response as well as identify any sex differences. Our study demonstrates that short-term consumption of a high-energy diet elicits notable sex-specific differences in body weight, body composition, inflammatory markers, gut microbiota, and fat excretion in Sprague-Dawley rats. These findings emphasize the importance of considering sex as a biological variable in studies of diet-induced obesity and its metabolic consequences.

## 2. Materials and Methods

### 2.1. Animals

Male and female Sprague-Dawley rats, known to be prone to diet-induced obesity [37] (*n* = 6 per group; ~300 g; Envigo, Indianapolis, IN, USA), were housed individually in conventional polycarbonate shoe-box cages in a temperature-controlled vivarium with ad libitum access to low-energy-density (LED; 5% fat and 3.25% sucrose) pellets of rat chow (PicoLab rodent diet 20, product #5053, Fort Worth, TX, USA) and water. Rats were maintained on a 12:12 h light/dark cycle with lights on at 0700 h and allowed to acclimate to laboratory conditions for 1 week prior to starting experiments. All animal procedures were approved by the University of Georgia Institutional Animal Care and Use Committee and conformed to National Institutes of Health Guidelines for the Care and Use of Laboratory Animals.

### 2.2. Food Intake, Body Weight, and Body Composition

Following the acclimation period, rats were maintained on a low-energy-density (LED) diet for an additional ten days and were then switched to a high-energy-density (HED) diet (45% fat and 20% sucrose, Research Diet #D12451, New Brunswick, NJ, USA) for four weeks. Food intake was measured twice. Body weight and body composition were measured weekly using a minispec LF 110 BCA Analyzer (Bruker Corp., The Woodlands, TX, USA).

### 2.3. Cytokines, Leptin, and Insulin Levels in Serum

Blood samples were collected on the last day of the LED diet and the last day of the HED diet by cardiac puncture. The serum was collected and stored at −21 °C. A cytokine array (Rat Cytokine ELISA Kit, cat #EA-4006, Signosis Inc., Sunnyvale, CA, USA) was used to measure levels of cytokines and chemokines. Insulin levels were determined using the Rat Insulin ELISA kit (cat #80-INSRT-E01; ALPCO Diagnostics, Inc., Salem, NH, USA).

### 2.4. Microbiome Analysis

Fecal samples were collected at four timepoints: upon arrival at the animal facility, after ten days of the LED diet, after one week on the HED diet, and after four weeks on the HED diet. Bacterial DNA was extracted from feces using a commercial kit (Quick-DNA Fecal/Soil Microbe Miniprep Kit, cat #D6010, Zymo research, Irvine, CA, USA). High-throughput sequencing was performed using Illumina MiSeq paired-end runs (GGBC, Athens, GA, USA). The raw sequences were demultiplexed, filtered, trimmed, and denoised using the software Quantitative Insights Into Microbial Ecology 2 (QIIME2), version 2023.2 [38]. Subsequently, features were identified, and Operational Taxonomic Units (OTUs) were picked against the GreenGenes database for further analysis. Using the microbiome analyst platform, bacterial abundance was normalized by log-transformation, and β-diversity (dissimilarities and similarities between samples) was assessed via Principal Coordinates Analysis (PCoA), with distances determined using the Bray–Curtis index. Significant dissimilarities between groups were determined via permutational multivariate ANOVA (PERMANOVA) [39].

### 2.5. Lipid Content in Feces

Fecal samples were collected after four weeks of HED diet consumption and stored at −21 °C. Samples from aged-matched, LED diet-fed rats were used for comparison. The samples were processed and analyzed to determine the lipid content using the acid steatocrit method, as described by Tran et al. [40]. The stool (0.5 g) was diluted with a volume of deionized water equal to two times the weight of the stool and homogenized using a 5 mL Potter Elvehjem tissue homogenizer (Wheaton, Vineland, NJ. USA). Next, 5N Perchloric acid (HClO_4_) was added to the homogenate in a volume equal to one-fifth of the homogenate volume. The resulting homogenate was mixed for 30 s using a standard Vortex mixer (Fisher Scientific, Waltham, MA, USA). The homogenate was aspirated into a 75 micro-liter glass hematocrit capillary tube. The capillary tube was centrifuged horizontally at 13,000 rpm for 15 min. After centrifugation, the lengths of the upper fatty layer and the bottom solid layers were measured by means of a graduated magnifying lens. Steatocrit was calculated using the following equation:$$\frac{Fatty\;layer\;length}{Fatty\;layer\;length+Solid\;layer\;length}\times 100$$

### 2.6. Statistical Analysis

GraphPad Prism 7 (GraphPad Software, Inc., Boston, MA, USA) was used to conduct statistical analyses, except when noted otherwise. Data are expressed as mean ± SEM and were analyzed using *t*-test or ANOVA, followed by the Tukey multiple comparisons post hoc test, as appropriate, unless otherwise noted. The alpha value was set at 0.05. Caloric intake/body weight, body weight gain, and fat content in feces were analyzed using a two-way ANOVA, followed by the Tukey post hoc test, as it produces the narrowest confidence intervals for pairwise comparisons. Body composition and cytokine levels were analyzed using a two-way RM ANOVA, followed by the Tukey post hoc test, as it produces the narrowest confidence intervals for pairwise comparisons.

## 3. Results

### 3.1. High-Energy-Density Diet Consumption Significantly Increased Body Weight and Fat Mass

Caloric intake, body weight gain, and body composition are shown in Figure 1. Given that males and females have inherently different caloric needs due to differences in body size, caloric intake is shown as calories consumed divided by body weight in grams. Male and female rats significantly increased their caloric intake in the first week after the introduction of the HED diet compared to their intake on the LED diet (F2.071, 20.71 = 54.70, *p* < 0.0001) (Figure 1A). This increase was transient, and the caloric intake returned to LED diet levels by the second week on the HED diet and remained stable thereafter. Caloric intake was slightly higher in females than males during the second week of HED diet consumption (*p* < 0.05). Body weight was analyzed as body weight gain from the last day of the LED diet; within the first week of HED feeding, there was a noticeable variance within the males with some gaining more weight than others. Thus, the male cohort was divided into two groups based on mean body weight gain [41,42,43] for further analysis: DIO-P—males with the highest weight gain and DR—males with the lowest weight gain. Two-way ANOVA showed the main effect of time (F1, 9 = 48.06, and *p* < 0.0001) and cohort (F2, 9 = 18.31, and *p* < 0.001). All rats gained weight within a week of HED feeding (Figure 1B), with DIO-P rats gaining more weight than DR and female rats (*p* < 0.05). After four weeks on the HED diet, the weight gained nearly doubled compared to week one (*p* < 0.05). There was no difference between DIO-P and DR rats; however, all males gained significantly more weight than females (*p* < 0.01).

Consistent with significant weight gain, body fat increased significantly with HED diet consumption. Two-way ANOVA on fat mass gain showed the main effect of time (F1, 9 = 26.26, and *p* < 0.001) and cohort (F2, 9 = 18.13, and *p* < 0.001) (Figure 1C). Within each group, fat mass accumulation was significantly higher after four weeks on the HED diet compared to one week (*p* < 0.05). After one week of HED feeding, DIO-P rats gained significantly more fat mass than DR and female rats (*p* < 0.01). This difference became more pronounced after four weeks on the HED diet (*p* < 0.001). There were no differences between DR and female rats. All rats continued to grow, as shown by their lean mass (Figure 1D), although male rats were gaining significantly more lean mass than female rats (*p* < 0.05).

### 3.2. HED Diet Decreased Microbial Diversity Within a Week

The analysis of fecal microbiota composition is shown in Figure 2. Fecal samples were collected on day one of the LED diet, at the end of the LED diet, one week after the introduction of the HED (HED1) diet, and after four weeks on the HED (HED4) diet. There was no significant difference between the two LED diet timepoints; thus, the data were combined for those two points. We found no significant differences in fecal fat content between DR and DIO-P rats. Thus, the data are presented together. Data for males and females are presented separately to better appreciate sex differences. High-energy-density diet consumption significantly reduced the number of different bacterial species present in the gut microbiota, as evidenced by the Shannon index (two-way ANOVA, F(2,39) = 35.25, and *p* < 0.0001) in males and females within one week (Figure 2A). In males, a PERMANOVA analysis of the (dis-)similarities between samples revealed a clear separation between the fecal microbiota of the LED and HED diet timepoints (F = 6.321; *p* = 0.001) (Figure 2B), with no significant differences between DR and DIO-P rats. Pairwise analysis with multi-testing adjustment based on the Benjamini–Hochberg procedure reveals that microbiota composition while on the LED diet was significantly different from the HED1 diet (*p* < 0.05) and HED4 diet (*p* < 0.05). In females, a PERMANOVA analysis of (dis-)similarities between samples reveals a clear separation between the fecal microbiota of the LED and HED diet timepoints (F = 12.237; *p* = 0.001) (Figure 2B). Microbiota composition on the LED diet was significantly different from the HED1 diet (*p* = 0.003) and HED4 diet (*p* = 0.003). The microbiota composition of the HED1 diet was also significantly different from the HED4 diet (*p* = 0.005).

In general, HED diet consumption led to a significant decrease in species belonging to the Bacteroidetes phylum compared to the LED diet in males (two-way ANOVA, F(2,12) = 56.47, and *p* < 0.0001) and females (one-way ANOVA, F(2,21) = 25.86, and *p* < 0.0001) (Figure 2C). The abundance of members of the species caccae of the Bacteroides genus and distasonis of the Parabacteroides genus were significantly depleted by the HED diet compared to the LED diet (*p* < 0.01). Members of the uniformis species of the Bacteroides genus were significantly depleted only in females after four weeks of HED diet consumption. We further identified a significant increase in species belonging to the Firmicutes phylum in males (two-way ANOVA, F(2,12) = 66.76, and *p* < 0.0001) and females (one-way ANOVA, F(2,18) = 9.552, and *p* = 0.0015). Members of the Flavefaciens species of the Ruminococcus genus were significantly increased in DIO-P and female rats (*p* < 0.0001) after four weeks of HED diet intake. In addition, the Muciniphila species of the Akkermansia genus (Verrucomicrobia phylum) was significantly increased after four weeks of the HED diet in males (two-way ANOVA, F(2,36) = 110.7), and *p* < 0.0001) and females (one-way ANOVA, F(2,21) = 6.68, and *p* = 0.0057). The increase was significantly more pronounced in DR rats compared to DIO-P and female rats (*p* < 0.0001).

We performed a simple linear regression analysis between the identified taxa and the phenotypic outcomes assessed in this study. Several correlations were found to be significant and are shown in Figure 3. In males, there is a negative correlation between the abundance of the Caccae species and caloric intake (Slope = −0.051, and CI [−0.089 to −0.012]). There is a negative correlation between the abundance of the Muciniphila species (Slope = −0.0006, and CI [−0.001058 to −5.783 × 10^−5^]) and the Verrucomicrobia phylum (Slope = −0.5301 and CI [−0.9131 to −0.1470]) and percent fat mass. The abundance of the Distasonis species was negatively correlated with levels of IL−1β (Slope = −2.767 × 10^−5^, and CI [−4.656 × 10^−5^ to −8.769 × 10^−6^]). The abundance of members of the Firmicutes phylum was positively correlated with percent fat mass (Slope = 0.2870, and CI [0.001699 to 0.5722]). In females, abundance of the Flavefaciens species was negatively correlated with body weight (Slope = −0.0008, and CI [−0.001438 to −0.0001370]). There was a negative correlation between the abundance of the Caccae species and percent fat mass (Slope = −0.013, and CI [−0.02545 to −0.001003]). The abundance of members of the Verrucomicrobia phylum was positively associated with body weight (Slope = 2.349, and CI [0.4095 to 4.288]).

### 3.3. Male Rats Excrete More Fat in Their Feces than Female Rats

Analysis of fat content in fecal samples from the rats fed LED and HED diets reveals that males, independent of diet, excreted significantly more fat in their feces than females (F1, 19 = 42.45, and *p* < 0.0001) (Figure 4). The difference was evident in LF diet-fed animals and became more pronounced with HED diet consumption.

### 3.4. HED Diet Consumption Affects Cytokine and Chemokine Levels in Serum

Cytokine and chemokine levels were measured in sera collected at the end of the ten-day LED diet and after four weeks of HED diet consumption using an ELISA cytokine array and are shown in Figure 5 and Table 1. We observed that levels of IL-1α were significantly lower after being on the HED diet for four weeks (F(1,10) = 30.77, and *p* < 0.001) (Figure 5A). This effect was independent of sex. Consumption of the HED diet significantly increased TGF-β in male rats, but it was not affected in female rats (*p* < 0.05) (Figure 5B). In contrast, IP-10 was significantly increased by HED diet consumption in female rats but not in male rats (*p* < 0.05) (Figure 5C). Levels of RANTES were significantly decreased by the HED diet in female rats but remained unaffected in males (*p* < 0.05) (Figure 5D). FGFb was not affected by HED diet consumption (F(1,10) = 0.2210, *p* = 0.6484) (Figure 5E). However, HED diet-fed females had significantly higher levels than males (*p* < 0.05) (Figure 5E). Further analysis reveals that in LED diet-fed males, DR rats had significantly higher levels of FGFb compared to DIO-P rats (Figure 5F). This difference was abolished with HED diet consumption (Figure 5F). Consistent with previous reports, insulin was significantly higher after the HED diet compared to the LED diet (F(1,8) = 174.3, *p* < 0.0001) (Figure 5G), with males exhibiting significantly higher levels than females (*p* < 0.05) (Figure 5G). Furthermore, there was no difference in insulin within the male cohort when fed the LED diet. However, DIO-P rats had higher levels of insulin than DR rats (Figure 5H). Short-term HED diet consumption did not affect serum levels of IL-6, VEGF, TNF-α, IFN-γ, MCP-1, SCF, MIP-1α, IL-1β, IL-5, IL-15, and leptin (Table 1).

## 4. Discussion

This study investigated sex-specific differences in body weight, body composition, inflammatory markers, gut microbiome, and fecal fat excretion following short-term HED diet consumption in rats. Here, we show that although males and females exhibit an increase in body weight, fat mass accumulation, and fat excretion in feces in response to the HED diet, the magnitude of the response differs between the sexes. The immediate impact on the gut microbiota composition is varied and complex, as we identified the depletion of *uniformis* and *distasonis* species but also increased abundance of *flavefaciens* and *muciniphila* species, all of which have been associated with benefits to the host. We further report differences in inflammatory markers in males and females in response to HED diet consumption. The HED diet induced a significant decrease in IL-1α in all rats. Insulin levels were significantly increased by the HED diet in all rats. TGFb was significantly increased after the HED diet but only in males. Interferon gamma-inducible protein 10 and RANTES were significantly increased by the HED diet but only in females. Fat excretion in feces was significantly higher after the HED diet, and this response was more pronounced in males than females. This discrepancy may be explained by sex differences in lipid absorption and metabolism.

Consistent with prior reports, consumption of the HED diet triggered a transient increase in caloric intake that was quickly reduced to a level equivalent to the LED diet intake [13,44,45]. Although males are heavier than females [46] and thus have a greater caloric requirement [47], there was no difference in caloric intake between males and females after adjusting for body weight. Despite males and females having similar caloric intakes relative to body weight, male rats gained twice as much weight as females, 48.8 g vs. 20.7 g, respectively. Males were also quicker than females to show significant increases in body weight. These findings add to the existing body of literature suggesting that males are highly susceptible to HED diet-induced obesity, while females exhibit certain protection from short-term HED diet-induced alterations in energy balance [48,49,50]. In a study on the proteome of brown adipose tissue by Choi et al., female rats had greater levels of proteins associated with thermogenesis and fat oxidation than males, along with lower levels of lipogenic proteins [51].

A trend was observed when looking at body weight gain, as half of the males gained significantly more weight than the others. This is consistent with the DIO-P and DR phenotypes first identified by Levin et al. in 1983 [9]. Consistent with previous reports, when body weight gain was examined between these two phenotype groups separately, males in the DIO-P group had significantly greater body weight gain than DR rats [52,53]. Although there was no significant difference between the groups after the fourth week of the HED diet, this is likely due to the cohort size.

As previously shown [54,55,56], HED diet consumption induced an increase in body fat within one week of HED feeding, and body fat continued to increase until the end of the HED diet. Prior studies have reported a significantly higher body fat gain in males than in females [57,58], yet only males in the DIO-P cohort had greater fat mass gain than females. DR rats had a similar body fat gain as females despite having greater body weight gain than females. Males in the DIO-P group had a significantly higher body fat gain than the DR group after one week and four weeks on the HED diet, which is consistent with prior reports [13,37]. Concurrent with the increase in body weight in both males and females, both male cohorts and females had significantly increased lean mass at four weeks on the HED diet from the first week of the HED diet.

Gut bacterial diversity was significantly reduced by HED diet consumption. Low microbial diversity has been associated with adiposity, insulin resistance, and a more detrimental inflammatory profile [59]. Although the validity of the *Firmicutes* to *Bacteroidetes* ratio as a hallmark of the obese phenotype is still widely discussed, previous reports have shown that there is a decrease in *Bacteroidetes* abundance with a proportional increase in *Firmicutes* in obese compared to lean subjects [44,60,61]. Results from the present study further corroborate these findings, given that one week of HED diet consumption was sufficient to trigger a significant increase in members of the *Firmicutes* phylum and a significant decrease in members of the *Bacteroidetes* phylum. The abundance of members of the *Firmicutes* phylum was also positively associated with fat mass accumulation in male rats. We show that the HED diet decreased the abundance of the *uniformis* species, which is considered a beneficial bacterium as it can produce short-chain fatty acids and other metabolites that help regulate metabolism [62]. Although this finding did not reach statistical significance in males, this is likely due to the small sample size. Similarly, *Parabacteroides distasonis*, which was significantly reduced by the HED diet and negatively associated with levels of IL-1β in males, has been shown to improve the obese phenotype through the production of succinate and secondary bile acids in the gut [63,64]. Interestingly, we identified a decrease in the abundance of the *caccae* species. This change has potential beneficial consequences as this species has been associated with increased intestinal inflammation [65]. The observed increase in the abundance of *flavefaciens* and *muciniphila* species could represent an initial compensatory response from the host to counteract the detrimental metabolic impact of the HED diet since the *flavefaciens* species has been reported to be essential in reducing hepatic fat accumulation [66], and we identified a negative correlation between *flavefaciens* abundance and body weight in female rats. Interestingly, in males, the increase in abundance of the *flavefaciens* species was less pronounced in DR rats than in DIO-P rats. Further studies with larger sample sizes are necessary to explore the physiological significance of this response. The *muciniphila* species are known as beneficial microbes that have been associated with improved body fat mass and glucose tolerance [67], and we observed a negative correlation between *muciniphila* abundance and fat mass accumulation in male rats. The colonization of HF-fed mice with *A. muciniphila* improves metabolic profiles by reducing inflammation in chow diet-fed mice. In addition, glucose tolerance was significantly improved, and serum levels of leptin insulin and leptin were decreased [68]. *A. muciniphila* is thought to reduce inflammation in visceral adipose tissue and inhibit the release of proinflammatory cytokines in stromal vascular fraction cells [68]. In addition, it has been shown that these effects depend on live *A. muciniphila*, as treatment with heat-killed cells did not enhance the metabolic profile in HF diet-fed obese mice [69]. Furthermore, *A. muciniphila* has been shown to improve metabolic profile in animals fed a low-fat, regular chow diet [70]. Interestingly, the abundance of the *Verrucomicrobia* phylum was positively correlated with body weight in female rats and negatively correlated with fat mass percent in male rats.

Consistent with prior reports, HED diet consumption significantly increased fecal fat excretion in both males and females [71,72]. The fat content in feces nearly tripled in males and doubled in females after the HED diet compared to the LED diet. This discrepancy may be explained by sex differences in lipid absorption and metabolism. Female Sprague-Dawley rats have been reported to absorb dietary fat significantly faster and to be more efficient at oxidizing fat than males [73]. Slower fat absorption in males may explain the elevated fecal fat excretion, and increased oxidizing efficiency might remediate the increased fat absorption in females. Furthermore, estrogen in females has been hypothesized to influence sex differences in fat metabolism. Estrogen has been found to inhibit lipoprotein lipase, which hydrolyzes triglycerides into fatty acids for uptake by fat cells, and increase growth hormone, leading to lower fat storage and increasing fat mobilization [74].

Our results reveal that HED diet consumption induced significant changes in levels of IL-1α, TGF-β, IP-10, RANTES, and insulin. Both males and females had significantly decreased levels of IL-1α after four weeks of HED diet consumption. This finding is consistent with prior reports on males [44] and females [75]. A study using IL-1α KO mice suggests that a deficiency and/or reduction in IL-1α might exert a protective effect against HED diet-induced obesity and its metabolic consequences through a finding of decreased adipose tissue weight and glucose intolerance in the knockout mice [76].

HED diet consumption significantly increased levels of anti-inflammatory cytokine TGF-β in male rats, consistent with a report from Kundu et al. [77]. There were no changes in TGF-β in female rats. Pinhal et al. reported that female rats fed an HED diet did not have significantly increased renal expression of TGF-β after five weeks on the diet [78]. Although findings from a study using Smad3 knockout mice suggest that suppression of the TGF-β/Smad3 improves insulin sensitivity [79], the increase observed in male rats may be due to its ability to inhibit adipogenesis [80] and might be mediated by testosterone, as previous studies have shown that testosterone can increase plasma levels of TGF-β [81,82,83].

In the current study, HED diet consumption significantly increased IP-10 levels in females while having no effect on males. This is consistent with our previous report on a short-term HED diet in males [44]. Given that females have been previously shown to exhibit no changes in IP-10 levels in response to a 60% fat diet after a 10-month dietary intervention [84], it is possible that this immediate response disappears over time. The observed difference in responses between males and females may be partially mediated by estrogen. IP-10 induces the release of additional proinflammatory cytokines, and this response is attenuated in the presence of estrogen [85]. It is plausible that the elevated levels of IP-10 serve as a compensatory mechanism to ensure the proper magnitude of the inflammatory response.

Serum levels of RANTES were unaffected by HED feeding in males; however, we observed a significant decrease in females. Previous studies have reported an increase in the mRNA expression of RANTES in the pancreas [86] and muscle tissue [87] in male rats in response to long-term high-fat consumption, yet no response to short-term HED diet intake [44]. To our knowledge, this is the first time circulating levels of RANTES have been evaluated in female rats in response to HED diet consumption. However, a previous study in female rats exposed to a high-cholesterol diet revealed that estrogen suppressed diet-induced RANTES expression in hippocampal tissue [88].

The HED diet had no significant impact on FGFb levels in rats. Under LED diet conditions, male rats exhibited slightly lower levels than females. This difference became more pronounced on the HED diet, largely due to a slight decrease in FGFb levels in DR male rats. There is limited research on the involvement of FGFb in obesity development and progression in rodent models and human subjects. However, FGFb is known to induce strong vascularization and has been reported to induce the generation of new fat cells in mice [89]. It is possible that the effects of HED diet consumption on FGFb are tissue-specific rather than systemic, suggesting the need for further research.

Consistent with previous reports, the HED diet significantly increased insulin levels in all animals. However, this increase was significantly more pronounced in DIO-P males compared to females and DR males [37,90,91,92,93,94].

Although the HED diet triggered a significant decrease in IL-1α in both males and females, the other interleukin measured—IL-6, IL-5, IL-15, and IL-1β—were unaffected. In addition, there were no changes in levels of VEGF, TNF-α, MCP-1, IFN-γ, MIP-1α, and SCF.

Leptin is produced by white adipose tissue adipocytes and acts as a hormone in appetite suppression and as a proinflammatory cytokine in immune signaling through the upregulation of TNF-a and IL-6 secretion from macrophages [95,96]. In the current study, no differences in leptin levels were found. Males exhibiting no significant changes in leptin levels is consistent with our prior report and one by Gómez-Crisóstomo et al., in which males fed either a control, 30% sucrose, 30% fat, or 30% sucrose/30% fat diet did not show any significant differences in serum leptin after two months on the diets [44,97]. Females have also shown this trend in a four-week study on two HED diets, although the two diets contained 78.7% kcal of fat and 92.8% kcal of fat, respectively [98]. The lack of an effect on leptin levels between the diets may be explained by the ability of leptin to induce STAT3 signaling in short-term HED diets, which is diminished as the diet continues in the long term [99]. The hypothalamic STAT3 signaling pathway is required for leptin’s actions on food intake and becomes resistant to activation by leptin after prolonged high-fat feeding [99,100]. This short-term HED diet was not long enough to induce leptin resistance in the STAT3 pathway, thus the leptin levels were not elevated from the baseline.

## 5. Conclusions

Despite significant advancements in understanding how sex influences obesity pathophysiology, previous obesity research has frequently been gender-neutral or under-represented women. There is often a lack of sex-specific analysis. Additionally, although an increasing number of studies now include both sexes, differences in animal species, experimental design, length of dietary intervention, and outcomes measured warrant the need for further exploration of this topic in the field. In male mice fed an HED diet for five weeks, males exhibited an increase in caloric intake, body weight, and fat mass, while females only increased in body weight [49]. Female and male mice have also been shown to experience distinct changes in gut microbiota composition in response to high-fat feeding [101,102,103]. Furthermore, Maric et al. reported that when exposed to the same experimental paradigm, male and female rodents reacted quite differently to an HED diet challenge and utilized distinct compensatory energy expenditure mechanisms. Also, these sex differences vary between rats and mice [36]. Our study demonstrates that short-term consumption of a high-energy-density (HED) diet elicits notable sex-specific differences in body weight, body composition, inflammatory markers, gut microbiota, and fat excretion in Sprague-Dawley rats. While both males and females experienced increases in body weight, fat mass, and fecal fat excretion, males were more susceptible to significant weight gain, particularly in the diet-induced obesity-prone (DIO-P) group. Differences in inflammatory responses further highlight sex-specific adaptations, with males showing elevated TGF-β levels, while females exhibited increases in IP-10 and RANTES. Changes in gut microbiota composition, characterized by a reduction in beneficial species like *Bacteroides uniformis* and *Parabacteroides distasonis* and an increase in species such as *Akkermansia muciniphila*, suggest complex host–microbe interactions that may mitigate the metabolic impact of the HED diet. Additionally, sex differences in fat metabolism, likely influenced by estrogen, were evident in the greater fecal fat excretion observed in males. Our findings emphasize the need to consider sex differences in obesity research and dietary interventions.

Several limitations should be considered when interpreting the results of this study. First, the absence of a control (LED) diet cohort for the entire duration of the study limits the ability to compare long-term effects directly. Additionally, incorporating both the intraperitoneal glucose tolerance test and the insulin tolerance test would have provided a more comprehensive understanding of the metabolic changes associated with the interventions. We also acknowledge that including shorter-term timepoints might have offered further insights into more transient effects, which could be valuable for future research. Finally, the study’s small sample size and short-term intervention duration also present limitations, as these factors may affect the generalizability and robustness of the findings.

## Figures and Tables

**Figure 1 nutrients-17-01147-f001:**
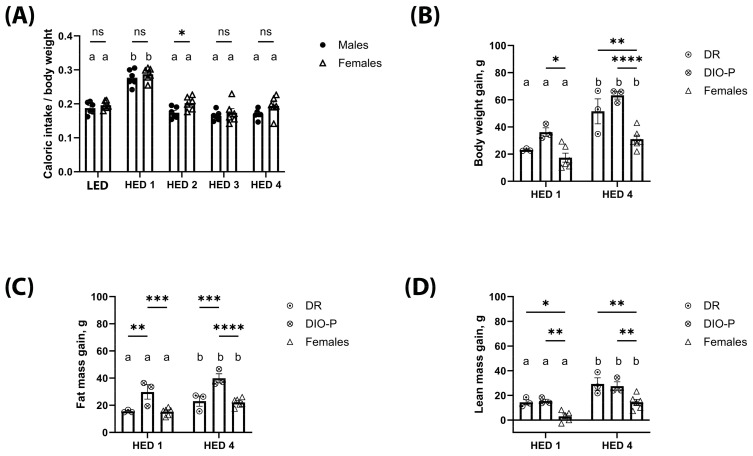
High-energy-density (HED) diet consumption significantly increased body weight and fat mass in male and female rats. The mean ± SEM kcal consumed per g body weight (**A**), body weight gain (**B**), fat mass gain (**C**), and lean mass gain (**D**) for male rats (n = 6) and female rats (n = 6) fed a baseline LED diet for ten days and then switched to HED diet for four weeks are shown. The male cohort was further split into DIO-P (n = 3) and DR (n = 3) per group. Both males and females significantly increased their caloric intake when switched to the HED diet (*p* < 0.01), but caloric intake decreased after one week and remained stable until the end of the study. All rats had significantly greater body weight gain after four weeks on HED diet than after one week of HED diet. Males gained significantly more body weight than females (*p* < 0.01). Males exhibited a clear 50% split in body weight gain after one week of HED feeding, which is consistent with DIO-P and DR phenotypes. DIO-P males had significantly greater body weight gain than females after one week of HED diet (*p* = 0.0116). All rats had significantly greater fat mass gain after four weeks on HED diet than after one week of HED diet (*p* < 0.05). DIO-P males had significantly greater fat mass gain than DR males and females within one week of HED feeding (*p* < 0.01). While all rats had significantly increased lean mass after four weeks on HED diet from the first week of HED diet, males in both cohorts had significantly increased lean mass gain than females at both timepoints (*p* < 0.05). Bars denoted with the same letter are not significantly different within each sex and between timepoints. Asterisks indicate statistical significance between sex or group at a given timepoint: * *p* < 0.05, ** *p* < 0.01, *** *p* < 0.001, and **** *p* < 0.0001. LED: low energy density; HED1 diet: high-energy-density diet, one-week consumption; HED4 diet: high-energy-density diet, four-week consumption.

**Figure 2 nutrients-17-01147-f002:**
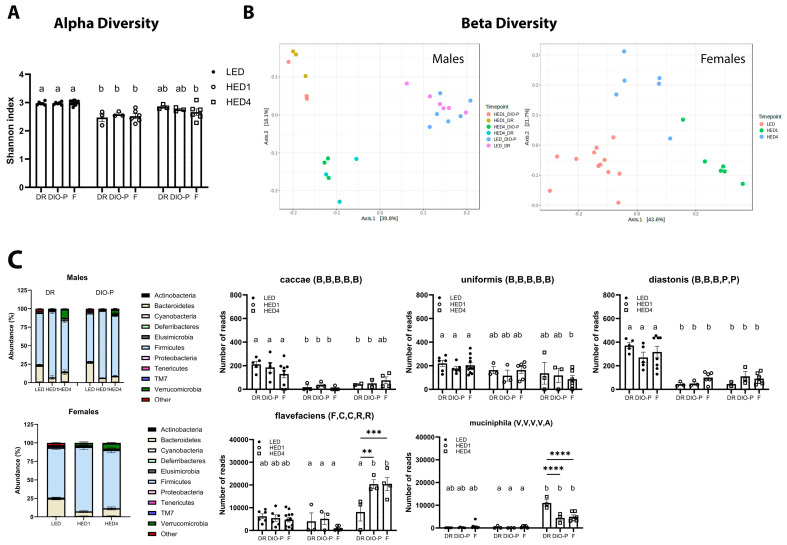
High-energy-density (HED) diet intake had a negative impact on gut microbiota composition. (**A**) Shannon index shown as mean + SEM for each group. HED diet significantly decreased the number of different bacterial species present in the gut microbiome within one week of diet introduction. (**B**) Principal coordinate analysis was analyzed using a pairwise PERMANOVA. The microbiota of the animals clustered together during LED diet and was significantly different after one week of HED diet and after four weeks of HED diet. (**C**) Bacterial abundance at the phylum level was quantified, and it is shown for males and females. Bacteroidetes and Firmicutes were the most abundant bacterial phyla in all groups. HED diet consumption significantly decreased the abundance of members of the Bacteroidetes phylum, specifically the caccae, uniformis, and distasonis species. HED diet consumption significantly increased members of the Firmicutes phylum, specifically the flavefaciens species, and the Verrucomicrobia phylum, specifically the muciniphila species. There were no differences in microbial composition or bacterial abundance between males and females. Bars denoted with the same letter are not statistically different within each sex and between timepoints. Asterisks indicate statistical significance between sexes at a given timepoint: ** *p* < 0.01, *** *p* < 0.001, **** *p* < 0.0001. LED (n = 12): low energy density; HED1 diet (n = 6): high-energy-density diet, one-week consumption; HED4 diet (n = 6): high-energy-density diet, four-week consumption; DR: diet-induced obesity resistant; DIO-P: diet-induced obesity prone. The letters in parenthesis next to species names refer to phylum, class, order, family, and genus that the species belongs to: B,B,B,B,B: p_Bacteroidetes, c_Bacteroidia, o_Bacteroidales, f_Bacteroidaceae, and g_Bacteroides; B,B,B,P,P: p_Bacteroidetes, c_Bacteroidia, o_Bacteroidales, f_Porphymonadaceae, and g_Parabacteroides; F,C,C,R,R: p_Firmicutes, c_Clostridia, o_Clostridiales, f_Ruminococcaceae, and g_Ruminococcus; V,V,V,V,A: p_Verrucomicrobia, c_Verrucomicrobiae, o_Verrucomicrobiales, f_Verrucomicrobiaceae, and g_Akkermansia.

**Figure 3 nutrients-17-01147-f003:**
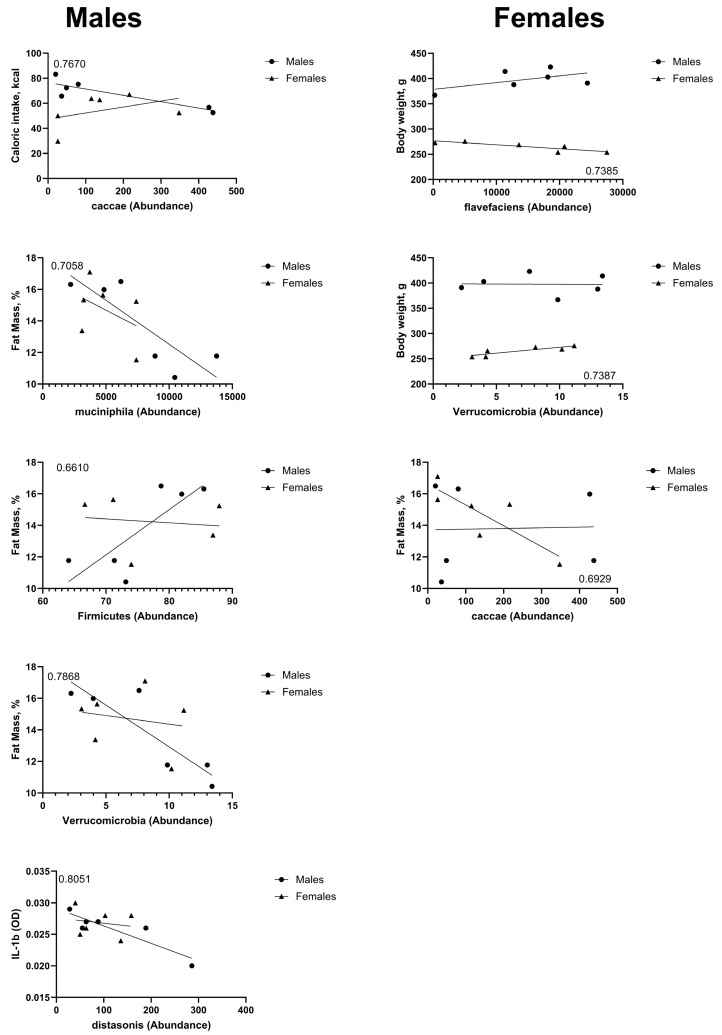
Bacterial taxa abundance after four weeks on HED diet was correlated with end-point caloric intake, body weight, percent fat mass, and levels of IL-1β. The results are shown, with accompanying R^2^ values, from simple linear regression analyses. Graphs on the left column represent correlations that were statistically significant (*p* < 0.05) in male rats; graphs on the right column represent correlations that were statistically significant (*p* < 0.05) in female rats.

**Figure 4 nutrients-17-01147-f004:**
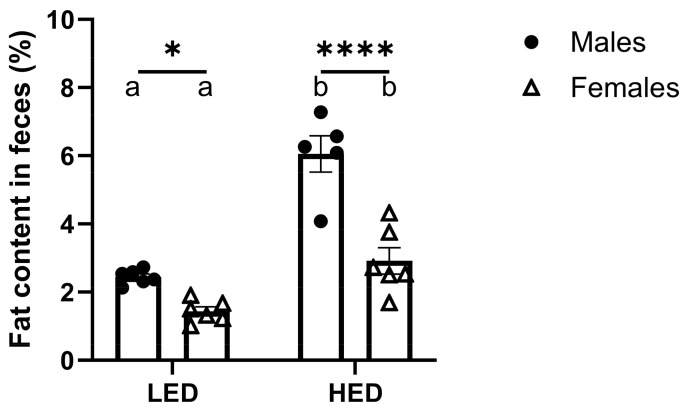
Consumption of a high-energy-density (HED) diet significantly increased fecal fat excretion in rats. The mean ± SEM fecal fat percentage for male rats (n = 6) and female rats (n = 6) is shown. HED diet consumption significantly increased fecal fat percentage in both males and females (Ps < 0.01), but males had a significantly greater fecal fat percentage than females after the LED (*p* = 0.0332) and HED diets (*p* < 0.0001). Bars denoted with the same letter are not statistically different within each sex and between timepoints. Asterisks indicate statistical significance between sexes at the given timepoint: * *p* < 0.05 and **** *p* < 0.0001. LED: low energy density; HED: high energy density.

**Figure 5 nutrients-17-01147-f005:**
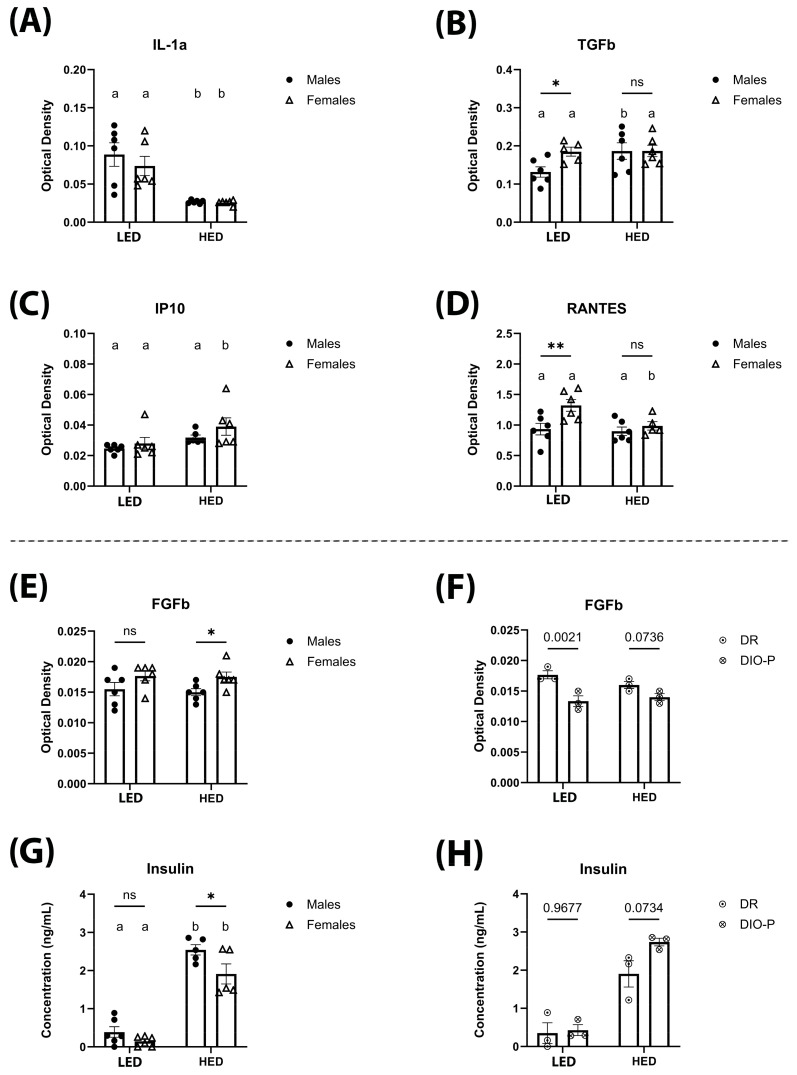
The mean ± SEM optical density (OD) of serum IL-1α (**A**), TGF-β (**B**), IP-10 (**C**), RANTES (**D**), FGFb (**E**), and FGFb for males split into DIO-P and DR groups (**F**) are shown. Serum concentration of insulin (ng/mL) is shown for males and females (**G**), and for males, it is split into DIO and DR groups (**H**). Both males and females had significantly decreased serum IL-1α after HED diet compared to LED diet (*p* = 0.0013 and 0.0065, respectively). Males had significantly increased levels of TGF-β after HED diet compared to LED diet (*p* = 0.0258), while females had significantly increased IP-10 levels after HED diet compared to LED diet (*p* = 0.0326). Females had significantly greater levels of RANTES than males after LED diet (*p* = 0.0037), which decreased significantly after HED diet (*p* = 0.0132). Females had significantly greater levels of FGFb than males after four weeks of the HED diet (*p* = 0.0478). When males were split into DIO-P and DR groups, males in the DR group had significantly higher levels of FGFb after LED diet (*p* = 0.0021), but there were no significant differences after HED diet. Both males and females had significant increases in serum insulin after HED diet (Ps < 0.0001), with males having significantly greater levels than females (*p* = 0.0148). Males in the DIO-P group had greater serum levels of insulin than males in the DR group after HED diet, but this difference was not significant (*p* = 0.0734). Bars denoted with the same letter within each sex are not significantly different. Asterisks indicate statistical significance between sexes or timepoints: * *p* < 0.05 and ** *p* < 0.01.

**Table 1 nutrients-17-01147-t001:** Cytokine levels in serum.

		Males		Females	
Cytokines	Category	LED	HED	LED	HED
IL-6	Pro- and anti-inflammatory	0.021 ± 0.001	0.027 ± 0.003	0.018 ± 0.001	0.020 ± 0.002
IL-5	Proinflammatory	0.059 ± 0.007	0.047 ± 0.003	0.053 ± 0.003	0.058 ± 0.005
IL-15	Proinflammatory	0.051 ± 0.006	0.045 ± 0.004	0.047 ± 0.003	0.046 ± 0.002
IL-1β	Proinflammatory	0.070 ± 0.007	0.056 ± 0.002	0.065 ± 0.003	0.062 ± 0.004
Leptin	Proinflammatory	0.050 ± 0.006	0.041 ± 0.002	0.046 ± 0.003	0.043 ± 0.003
VEGF	Proinflammatory	0.029 ± 0.003	0.027 ± 0.001	0.028 ± 0.002	0.028 ± 0.003
TNF-α	Proinflammatory	0.057 ± 0.007	0.044 ± 0.004	0.053 ± 0.003	0.050 ± 0.005
MCP-1	Proinflammatory	0.027 ± 0.003	0.022 ± 0.001	0.028 ± 0.003	0.023 ± 0.001
IFN-γ	Proinflammatory	0.056 ± 0.008	0.045 ± 0.004	0.052 ± 0.003	0.047 ± 0.003
MIP-1α	Proinflammatory	0.019 ± 0.001	0.017 ± 0.002	0.017 ± 0.001	0.020 ± 0.001
SCF	Stem cell factor	0.074 ± 0.010	0.076 ± 0.016	0.067 ± 0.004	0.058 ± 0.004

Values are expressed as means ± SEM. Cytokine levels were measured in male (n = 6) and female (n = 6) rats after ten days on a low-energy-density (LED) diet and after four weeks on a high-energy-density (HED) diet. Cytokine levels were measured using Optical Density (OD) of sera in a cytokine array.

## Data Availability

All raw data can be assessed at Raw Data.

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
