# Peer review of "Sex-Specific Effect of a High-Energy Diet on Body Composition, Gut Microbiota, and Inflammatory Markers in Rats"

_nutrients, 2025, doi:10.3390/nu17071147_

Round 1
Reviewer 1 Report
Comments and Suggestions for Authors
Review of “Differential Effects of High-Energy Diet Consumption on Body Composition, Gut Microbiota, and Inflammatory Markers in Male and Female Rats” (Submission ID nutrients-3488569)
This study investigated the differential effects of HED on food intake, body composition, gut microbiota, lipid content in feces and inflammatory markers in male and female Sprague-Dawley rats. This study demonstrated that short-term consumption of a high-energy diet shows sex-specific differences in body weight, body composition, inflammatory markers, gut microbiota, and fat excretion. This study is potentially interesting: however, several problems to be solved.
- Why the authors chose Sprague-Dawley rats for this study?
- Criteria to distinguish between DIO-P and DR need to be clarified in Methos section.
- A control group that continues to eat LED is needed.
- DIO-P and DO should be analyzed separately for intestinal bacteria, fecal fat excretion and inflammatory markers.
- The analysis methods used in the various analyses should be described in the statistical methods.
- It is unclear which timing samples are retrieved for the data in Figure 3.
- It is desirable to perform intraperitoneal glucose tolerance test and Insulin tolerance test.
Author Response
We thank the Reviewer for the comments, we have addressed them to the best of our abilities and believe the manuscript have been significantly improved. Our response to each specific comment is presented below. In addition, revisions to the manuscript have been highlighted in yellow so editors and reviewers can readily see changes made.
This study investigated the differential effects of HED on food intake, body composition, gut microbiota, lipid content in feces and inflammatory markers in male and female Sprague-Dawley rats. This study demonstrated that short-term consumption of a high-energy diet shows sex-specific differences in body weight, body composition, inflammatory markers, gut microbiota, and fat excretion. This study is potentially interesting: however, several problems to be solved.
Why the authors chose Sprague-Dawley rats for this study?
Response:
Sprague-Dawley rats were selected for this study as this strain is known to develop obesity when exposed to a high fat-high sugar diet [1], which makes them ideal subjects to study the mechanisms behind diet-induced obesity and related metabolic disorders.
Criteria to distinguish between DIO-P and DR need to be clarified in Methos section.
Response:
The following text was added to clarify the criteria to divide the male cohort into DIO-P and DR: “Thus, the male cohort was divided into two groups based on mean body weight gain for further analysis: DIO-P –males with highest weight gain– and DR –males with lowest weight gain.”
A control group that continues to eat LED is needed.
Response:
Thank you for your valuable feedback. We acknowledge that including a cohort on the control diet would have been ideal and will be included in our future studies in the discussion section.
DIO-P and DO should be analyzed separately for intestinal bacteria, fecal fat excretion and inflammatory markers.
Response:
We analyzed DIO-P and DIO animals separately, and found no significant differences in microbiota composition or fecal fat. Thus, the data are presented together. For inflammatory makers, we found significant differences between DIO-P and DIO rats in levels of FGFb and insulin, shown in Figure 5, F and H, respectively.
The analysis methods used in the various analyses should be described in the statistical methods.
Response:
The statistical analysis section has been revised to provide sufficient details.
It is unclear which timing samples are retrieved for the data in Figure 3.
Response:
We appreciate the attention to detail. The legend for Figure 3 has been revised as follows: “Bacterial taxa abundance after four weeks on HED was correlated with end-point caloric intake, body weight, percent fat mass, and levels of IL-1β.”
It is desirable to perform intraperitoneal glucose tolerance test and Insulin tolerance test.
Response:
Thank you for your suggestion. We agree that performing both the intraperitoneal glucose tolerance test and the insulin tolerance test would have been valuable additions to the study and this has been added to the discussion section as a limitation of our study.
References
[1] Levin BE, Dunn-Meynell AA, Balkan B, Keesey RE. Selective breeding for diet-induced obesity and resistance in sprague-dawley rats. Am J Physiol 1997;273:R725-30. doi: 10.1152/ajpregu.1997.273.2.R725
[2] Minaya DM, Turlej A, Joshi A, Nagy T, Weinstein N, DiLorenzo P, et al. Consumption of a high energy density diet triggers microbiota dysbiosis, hepatic lipidosis, and microglia activation in the nucleus of the solitary tract in rats. Nutr Diabetes 2020;10. doi: ARTN 20
10.1038/s41387-020-0119-4
[3] Shin NR, Lee JC, Lee HY, Kim MS, Whon TW, Lee MS, Bae JW. An increase in the akkermansia spp. Population induced by metformin treatment improves glucose homeostasis in diet-induced obese mice. Gut 2014;63:727-35. doi: 10.1136/gutjnl-2012-303839
[4] Everard A, Belzer C, Geurts L, Ouwerkerk JP, Druart C, Bindels LB, et al. Cross-talk between akkermansia muciniphila and intestinal epithelium controls diet-induced obesity. Proceedings of the national academy of sciences 2013;110:9066-71.
[5] Zhao S, Liu W, Wang J, Shi J, Sun Y, Wang W, et al. Akkermansia muciniphila improves metabolic profiles by reducing inflammation in chow diet-fed mice. J Mol Endocrinol 2017;58:1-14. doi: 10.1530/JME-16-0054
[6] Mendes LO, Scarano WR, Rochel-Maia SS, Fioruci-Fontaneli BA, Chuffa LG, Martinez FE. Testosterone therapy differently regulates the anti- and pro-inflammatory cytokines in the plasma and prostate of rats submitted to chronic ethanol consumption (uchb). Am J Reprod Immunol 2014;72:317-25. doi: 10.1111/aji.12268
[7] Liva SM, Voskuhl RR. Testosterone acts directly on cd4+ t lymphocytes to increase il-10 production. J Immunol 2001;167:2060-7. doi: 10.4049/jimmunol.167.4.2060
[8] De Paula GC, Simões RF, Garcia-Serrano AM, Duarte JM. High-fat and high-sucrose diet-induced hypothalamic inflammation shows sex specific features in mice. Neurochemical Research 2024;49:3356-66.
[9] Lewis DK, Bake S, Thomas K, Jezierski MK, Sohrabji F. A high cholesterol diet elevates hippocampal cytokine expression in an age and estrogen-dependent manner in female rats. J Neuroimmunol 2010;223:31-8. doi: 10.1016/j.jneuroim.2010.03.024
[10] Aronica SM, Fanti P, Kaminskaya K, Gibbs K, Raiber L, Nazareth M, et al. Estrogen disrupts chemokine-mediated chemokine release from mammary cells: Implications for the interplay between estrogen and ip-10 in the regulation of mammary tumor formation. Breast Cancer Res Tr 2004;84:235-45. doi: DOI 10.1023/B:BREA.0000019961.59306.f6
[11] Sutoyo DA, Atmaka DR, Sidabutar L. Dietary factors affecting firmicutes and bacteroidetes ratio in solving obesity problem: A literature review. Media Gizi Indones 2020;15:94-109.
[12] Magne F, Gotteland M, Gauthier L, Zazueta A, Pesoa S, Navarrete P, Balamurugan R. The firmicutes/bacteroidetes ratio: A relevant marker of gut dysbiosis in obese patients? Nutrients 2020;12. doi: 10.3390/nu12051474
[13] Priego T, Sanchez J, Pico C, Palou A. Sex-differential expression of metabolism-related genes in response to a high-fat diet. Obesity (Silver Spring) 2008;16:819-26. doi: 10.1038/oby.2007.117
[14] Eckel LA, Moore SR. Diet-induced hyperphagia in the rat is influenced by sex and exercise. Am J Physiol-Reg I 2004;287:R1080-R5. doi: 10.1152/ajpregu.00424.2004
[15] Taraschenko OD, Maisonneuve IM, Glick SD. Resistance of male sprague–dawley rats to sucrose-induced obesity: Effects of 18-methoxycoronaridine. Physiol Behav 2011;102:126-31.
[16] Roca P, Rodriguez AM, Oliver P, Bonet ML, Quevedo S, Picó C, Palou A. Brown adipose tissue response to cafeteria diet-feeding involves induction of the ucp2 gene and is impaired in female rats as compared to males. Pflügers Archiv 1999;438:628-34.
[17] Stubbins RE, Holcomb VB, Hong J, Núñez NP. Estrogen modulates abdominal adiposity and protects female mice from obesity and impaired glucose tolerance. European journal of nutrition 2012;51:861-70.
[18] Salinero AE, Anderson BM, Zuloaga KL. Sex differences in the metabolic effects of diet-induced obesity vary by age of onset. International journal of obesity 2018;42:1088-91.
[19] Yang Y, Smith Jr DL, Keating KD, Allison DB, Nagy TR. Variations in body weight, food intake and body composition after long‐term high‐fat diet feeding in c57bl/6j mice. Obesity 2014;22:2147-55.
[20] Asarian L, Geary N. Sex differences in the physiology of eating. American Journal of Physiology-Regulatory, Integrative and Comparative Physiology 2013;305:R1215-R67.
[21] Marco A, Schroeder M, Weller A. Microstructural pattern of palatable food intake from weaning to adulthood in male and female oletf rats. Behavioral neuroscience 2009;123:1251.
[22] Strohmayer A, Smith G. The meal pattern of genetically obese (ob/ob) mice. Appetite 1987;8:111-23.
[23] Kim JS, Kirkland RA, Lee SH, Cawthon CR, Rzepka KW, Minaya DM, et al. Gut microbiota composition modulates inflammation and structure of the vagal afferent pathway. Physiol Behav 2020;225. doi: ARTN 113082
10.1016/j.physbeh.2020.113082
[24] Minaya DM, Robertson KL, Rowland NE. Circadian and economic factors affect food acquisition in rats restricted to discrete feeding opportunities. Physiol Behav 2017;181:10-5. doi: 10.1016/j.physbeh.2017.09.003
[25] de Lartigue G, Barbier de la Serre C, Espero E, Lee J, Raybould HE. Leptin resistance in vagal afferent neurons inhibits cholecystokinin signaling and satiation in diet induced obese rats. Plos One 2012;7:e32967. doi: 10.1371/journal.pone.0032967
[26] Tabata Y, Miyao M, Inamoto T, Ishii T, Hirano Y, Yamaoki Y, Ikada Y. De novo formation of adipose tissue by controlled release of basic fibroblast growth factor. Tissue Eng 2000;6:279-89. doi: 10.1089/10763270050044452
Reviewer 2 Report
Comments and Suggestions for Authors
The authors are attempting to decipher how a high-calorie diet influences the physiology of both genders. To this end, they initiated a preclinical experiment using Sprague-Dawley rats of both genders. The laboratory animals were fed either a high-energy diet (HED), a low-energy diet (LED), or a combination of both.
Biochemical analyses of these subjects revealed that male mice fed an HED for five weeks exhibited an increase in body weight and fat mass, whereas females showed only an increase in body weight. Furthermore, both male and female mice displayed distinct changes in gut microbiota composition in response to the high-fat diet. The study demonstrates that long-term exposure to a high-energy diet leads to notable sex-specific differences in body weight, body composition, inflammatory markers, gut microbiota, and fat excretion in Sprague-Dawley rats.
The differences in inflammatory responses further highlight gender-specific adaptations, with males exhibiting slightly elevated levels of TGF-β, while females showed increased levels of IP-10, RANTES, and FGFb. The changes in gut microbiota composition, characterised by a reduction in beneficial species such as Bacteroides uniformis and Parabacteroides distasonis, along with an increase in Akkermansia muciniphila, suggest complex host-microbe interactions that may mitigate the impact of an HED.
This article underscores the importance of gender as a biological variable, which must be considered when examining diet-induced obesity and its metabolic consequences. This represents the novelty of the study, and the preliminary results presented in this manuscript merit publication.
Minor Observations:
1)A sentence cannot begin with an abbreviation. The authors must carefully review the manuscript and revise all such instances.
2)The manuscript contains numerous abbreviations, which are difficult to follow. For this reason, the authors must provide a list of abbreviations at the end of the manuscript, where they are properly explained.
Author Response
We thank the Reviewers for the comments, we have addressed them to the best of our abilities and believe the manuscript have been significantly improved. Our response to each specific comment is presented below. In addition, revisions to the manuscript have been highlighted in yellow so editors and reviewers can readily see changes made.
The authors are attempting to decipher how a high-calorie diet influences the physiology of both genders. To this end, they initiated a preclinical experiment using Sprague-Dawley rats of both genders. The laboratory animals were fed either a high-energy diet (HED), a low-energy diet (LED), or a combination of both.
Biochemical analyses of these subjects revealed that male mice fed an HED for five weeks exhibited an increase in body weight and fat mass, whereas females showed only an increase in body weight. Furthermore, both male and female mice displayed distinct changes in gut microbiota composition in response to the high-fat diet. The study demonstrates that long-term exposure to a high-energy diet leads to notable sex-specific differences in body weight, body composition, inflammatory markers, gut microbiota, and fat excretion in Sprague-Dawley rats.
The differences in inflammatory responses further highlight gender-specific adaptations, with males exhibiting slightly elevated levels of TGF-β, while females showed increased levels of IP-10, RANTES, and FGFb. The changes in gut microbiota composition, characterised by a reduction in beneficial species such as Bacteroides uniformis and Parabacteroides distasonis, along with an increase in Akkermansia muciniphila, suggest complex host-microbe interactions that may mitigate the impact of an HED.
This article underscores the importance of gender as a biological variable, which must be considered when examining diet-induced obesity and its metabolic consequences. This represents the novelty of the study, and the preliminary results presented in this manuscript merit publication.
Minor Observations:
1)A sentence cannot begin with an abbreviation. The authors must carefully review the manuscript and revise all such instances.
Response:
We appreciate the observation. The manuscript has been revised accordingly.
2)The manuscript contains numerous abbreviations, which are difficult to follow. For this reason, the authors must provide a list of abbreviations at the end of the manuscript, where they are properly explained.
Response:
A list of abbreviations has been included at the end of the manuscript.
References
[1] Levin BE, Dunn-Meynell AA, Balkan B, Keesey RE. Selective breeding for diet-induced obesity and resistance in sprague-dawley rats. Am J Physiol 1997;273:R725-30. doi: 10.1152/ajpregu.1997.273.2.R725
[2] Minaya DM, Turlej A, Joshi A, Nagy T, Weinstein N, DiLorenzo P, et al. Consumption of a high energy density diet triggers microbiota dysbiosis, hepatic lipidosis, and microglia activation in the nucleus of the solitary tract in rats. Nutr Diabetes 2020;10. doi: ARTN 20
10.1038/s41387-020-0119-4
[3] Shin NR, Lee JC, Lee HY, Kim MS, Whon TW, Lee MS, Bae JW. An increase in the akkermansia spp. Population induced by metformin treatment improves glucose homeostasis in diet-induced obese mice. Gut 2014;63:727-35. doi: 10.1136/gutjnl-2012-303839
[4] Everard A, Belzer C, Geurts L, Ouwerkerk JP, Druart C, Bindels LB, et al. Cross-talk between akkermansia muciniphila and intestinal epithelium controls diet-induced obesity. Proceedings of the national academy of sciences 2013;110:9066-71.
[5] Zhao S, Liu W, Wang J, Shi J, Sun Y, Wang W, et al. Akkermansia muciniphila improves metabolic profiles by reducing inflammation in chow diet-fed mice. J Mol Endocrinol 2017;58:1-14. doi: 10.1530/JME-16-0054
[6] Mendes LO, Scarano WR, Rochel-Maia SS, Fioruci-Fontaneli BA, Chuffa LG, Martinez FE. Testosterone therapy differently regulates the anti- and pro-inflammatory cytokines in the plasma and prostate of rats submitted to chronic ethanol consumption (uchb). Am J Reprod Immunol 2014;72:317-25. doi: 10.1111/aji.12268
[7] Liva SM, Voskuhl RR. Testosterone acts directly on cd4+ t lymphocytes to increase il-10 production. J Immunol 2001;167:2060-7. doi: 10.4049/jimmunol.167.4.2060
[8] De Paula GC, Simões RF, Garcia-Serrano AM, Duarte JM. High-fat and high-sucrose diet-induced hypothalamic inflammation shows sex specific features in mice. Neurochemical Research 2024;49:3356-66.
[9] Lewis DK, Bake S, Thomas K, Jezierski MK, Sohrabji F. A high cholesterol diet elevates hippocampal cytokine expression in an age and estrogen-dependent manner in female rats. J Neuroimmunol 2010;223:31-8. doi: 10.1016/j.jneuroim.2010.03.024
[10] Aronica SM, Fanti P, Kaminskaya K, Gibbs K, Raiber L, Nazareth M, et al. Estrogen disrupts chemokine-mediated chemokine release from mammary cells: Implications for the interplay between estrogen and ip-10 in the regulation of mammary tumor formation. Breast Cancer Res Tr 2004;84:235-45. doi: DOI 10.1023/B:BREA.0000019961.59306.f6
[11] Sutoyo DA, Atmaka DR, Sidabutar L. Dietary factors affecting firmicutes and bacteroidetes ratio in solving obesity problem: A literature review. Media Gizi Indones 2020;15:94-109.
[12] Magne F, Gotteland M, Gauthier L, Zazueta A, Pesoa S, Navarrete P, Balamurugan R. The firmicutes/bacteroidetes ratio: A relevant marker of gut dysbiosis in obese patients? Nutrients 2020;12. doi: 10.3390/nu12051474
[13] Priego T, Sanchez J, Pico C, Palou A. Sex-differential expression of metabolism-related genes in response to a high-fat diet. Obesity (Silver Spring) 2008;16:819-26. doi: 10.1038/oby.2007.117
[14] Eckel LA, Moore SR. Diet-induced hyperphagia in the rat is influenced by sex and exercise. Am J Physiol-Reg I 2004;287:R1080-R5. doi: 10.1152/ajpregu.00424.2004
[15] Taraschenko OD, Maisonneuve IM, Glick SD. Resistance of male sprague–dawley rats to sucrose-induced obesity: Effects of 18-methoxycoronaridine. Physiol Behav 2011;102:126-31.
[16] Roca P, Rodriguez AM, Oliver P, Bonet ML, Quevedo S, Picó C, Palou A. Brown adipose tissue response to cafeteria diet-feeding involves induction of the ucp2 gene and is impaired in female rats as compared to males. Pflügers Archiv 1999;438:628-34.
[17] Stubbins RE, Holcomb VB, Hong J, Núñez NP. Estrogen modulates abdominal adiposity and protects female mice from obesity and impaired glucose tolerance. European journal of nutrition 2012;51:861-70.
[18] Salinero AE, Anderson BM, Zuloaga KL. Sex differences in the metabolic effects of diet-induced obesity vary by age of onset. International journal of obesity 2018;42:1088-91.
[19] Yang Y, Smith Jr DL, Keating KD, Allison DB, Nagy TR. Variations in body weight, food intake and body composition after long‐term high‐fat diet feeding in c57bl/6j mice. Obesity 2014;22:2147-55.
[20] Asarian L, Geary N. Sex differences in the physiology of eating. American Journal of Physiology-Regulatory, Integrative and Comparative Physiology 2013;305:R1215-R67.
[21] Marco A, Schroeder M, Weller A. Microstructural pattern of palatable food intake from weaning to adulthood in male and female oletf rats. Behavioral neuroscience 2009;123:1251.
[22] Strohmayer A, Smith G. The meal pattern of genetically obese (ob/ob) mice. Appetite 1987;8:111-23.
[23] Kim JS, Kirkland RA, Lee SH, Cawthon CR, Rzepka KW, Minaya DM, et al. Gut microbiota composition modulates inflammation and structure of the vagal afferent pathway. Physiol Behav 2020;225. doi: ARTN 113082
10.1016/j.physbeh.2020.113082
[24] Minaya DM, Robertson KL, Rowland NE. Circadian and economic factors affect food acquisition in rats restricted to discrete feeding opportunities. Physiol Behav 2017;181:10-5. doi: 10.1016/j.physbeh.2017.09.003
[25] de Lartigue G, Barbier de la Serre C, Espero E, Lee J, Raybould HE. Leptin resistance in vagal afferent neurons inhibits cholecystokinin signaling and satiation in diet induced obese rats. Plos One 2012;7:e32967. doi: 10.1371/journal.pone.0032967
[26] Tabata Y, Miyao M, Inamoto T, Ishii T, Hirano Y, Yamaoki Y, Ikada Y. De novo formation of adipose tissue by controlled release of basic fibroblast growth factor. Tissue Eng 2000;6:279-89. doi: 10.1089/10763270050044452
Reviewer 3 Report
Comments and Suggestions for Authors
Major:
- The authors should clarify the selection of time points for microbiome and cytokine analyses. Were shorter-term effects considered?
- Regression analysis results require more details—provide confidence intervals for correlation coefficients.
- The discussion of Akkermansia muciniphila should be expanded to explain how its increase may be a compensatory response to gut dysbiosis rather than purely beneficial.
- Authors should discuss why TGF-β increased only in males, while IP-10 and RANTES increased in females.
Minor:
1. Authors should briefly mention the study limitations in the abstract.
Comments on the Quality of English LanguageMinor linguistic and stylistic corrections are required.
Author Response
We thank the Reviewer for the comments, we have addressed them to the best of our abilities and believe the manuscript have been significantly improved. Our response to each specific comment is presented below (in blue). In addition, revisions to the manuscript have been highlighted in yellow so editors and reviewers can readily see changes made.
The authors should clarify the selection of time points for microbiome and cytokine analyses. Were shorter-term effects considered?
Response:
Thank you for your comment. The selection of time points for the microbiome and cytokine analyses was based on our goal of capturing both immediate and progressive effects of the high-fat diet intake. While we did not include shorter-term time points in this study, we focused on those that would provide insight into both early and longer-term changes based on prior reports of gut microbiota modulation in high-fat fed rats [2]. We acknowledge that including shorter-term time points might have provided additional insight on more transient effects, and this could be an important consideration for future studies.
Regression analysis results require more details—provide confidence intervals for correlation coefficients.
Response:
We appreciate the recommendation. For all significant correlations, we have included the estimated slope value and 95% confidence interval.
The discussion of Akkermansia muciniphila should be expanded to explain how its increase may be a compensatory response to gut dysbiosis rather than purely beneficial.
Response:
We appreciate the recommendation. The following text has been added to the discussion section on A. muciniphila:
“Colonization of HF-fed mice with A. muciniphila improves metabolic profiles by reducing inflammation in chow diet-fed mice. Glucose tolerance was significantly improved and serum levels of insulin and leptin were decreased [3]. A. muciniphila is thought to reduce inflammation in visceral adipose tissue and inhibit the release of proinflammatory cytokines in stromal vascular fraction cells [3]. In addition, it has been shown that these effects depend on live A. muciniphila, as treatment with heat-killed cells did not enhance the metabolic profile in HF-fed obese mice [4]. Furthermore, A. muciniphila has been shown to improve metabolic profile in animals fed a low-fat, regular chow diet[5].”
Authors should discuss why TGF-β increased only in males, while IP-10 and RANTES increased in females.
Response:
We appreciate the recommendation. The discussion section has been modified as follows:
TGF-β: The following text has been added to the discussion section:
“…..might be mediated by testosterone as previous studies have shown that testosterone can increase plasma levels of TGF-β [6-8].”
RANTES: We observed a significant decrease in RANTES levels in females following the HED diet. The following text has been added to the discussion section:
“However, a previous study in female rats exposed to a high cholesterol diet revealed that estrogen suppressed diet-induced RANTES expression in hippocampal tissue [9].”
IP-10: The following text has been added to the discussion section:
“The observed difference in responses between males and females may be partially mediated by estrogen. IP-10 induces the release of additional proinflammatory cytokines, and this response is attenuated in the presence of estrogen [10]. It is plausible that the elevated levels of IP-10 serve as a compensatory mechanism to ensure the proper magnitude of the inflammatory response.”
Minor:
Authors should briefly mention the study limitations in the abstract.
Response:
We appreciate the recommendation. The following text has been added to the abstract:
“While we recognize that the study had a small sample size and a short-term intervention, our findings highlight the critical role of sex as a biological variable in diet-induced obesity research.”
In addition, we have included the following text to the conclusions section:
“Several limitations should be considered when interpreting the results of this study. First, the absence of a control diet cohort for the entire duration of the study limits the ability to compare long-term effects directly. Additionally, incorporating both the intraperitoneal glucose tolerance test and the insulin tolerance test would have provided a more comprehensive understanding of the metabolic changes associated with the interventions. We also acknowledge that including shorter-term time points might have offered further insights into more transient effects, which could be valuable for future research. Finally, the study's small sample size and short-term intervention duration also present limitations, as these factors may affect the generalizability and robustness of the findings.”
References
[1] Levin BE, Dunn-Meynell AA, Balkan B, Keesey RE. Selective breeding for diet-induced obesity and resistance in sprague-dawley rats. Am J Physiol 1997;273:R725-30. doi: 10.1152/ajpregu.1997.273.2.R725
[2] Minaya DM, Turlej A, Joshi A, Nagy T, Weinstein N, DiLorenzo P, et al. Consumption of a high energy density diet triggers microbiota dysbiosis, hepatic lipidosis, and microglia activation in the nucleus of the solitary tract in rats. Nutr Diabetes 2020;10. doi: ARTN 20
10.1038/s41387-020-0119-4
[3] Shin NR, Lee JC, Lee HY, Kim MS, Whon TW, Lee MS, Bae JW. An increase in the akkermansia spp. Population induced by metformin treatment improves glucose homeostasis in diet-induced obese mice. Gut 2014;63:727-35. doi: 10.1136/gutjnl-2012-303839
[4] Everard A, Belzer C, Geurts L, Ouwerkerk JP, Druart C, Bindels LB, et al. Cross-talk between akkermansia muciniphila and intestinal epithelium controls diet-induced obesity. Proceedings of the national academy of sciences 2013;110:9066-71.
[5] Zhao S, Liu W, Wang J, Shi J, Sun Y, Wang W, et al. Akkermansia muciniphila improves metabolic profiles by reducing inflammation in chow diet-fed mice. J Mol Endocrinol 2017;58:1-14. doi: 10.1530/JME-16-0054
[6] Mendes LO, Scarano WR, Rochel-Maia SS, Fioruci-Fontaneli BA, Chuffa LG, Martinez FE. Testosterone therapy differently regulates the anti- and pro-inflammatory cytokines in the plasma and prostate of rats submitted to chronic ethanol consumption (uchb). Am J Reprod Immunol 2014;72:317-25. doi: 10.1111/aji.12268
[7] Liva SM, Voskuhl RR. Testosterone acts directly on cd4+ t lymphocytes to increase il-10 production. J Immunol 2001;167:2060-7. doi: 10.4049/jimmunol.167.4.2060
[8] De Paula GC, Simões RF, Garcia-Serrano AM, Duarte JM. High-fat and high-sucrose diet-induced hypothalamic inflammation shows sex specific features in mice. Neurochemical Research 2024;49:3356-66.
[9] Lewis DK, Bake S, Thomas K, Jezierski MK, Sohrabji F. A high cholesterol diet elevates hippocampal cytokine expression in an age and estrogen-dependent manner in female rats. J Neuroimmunol 2010;223:31-8. doi: 10.1016/j.jneuroim.2010.03.024
[10] Aronica SM, Fanti P, Kaminskaya K, Gibbs K, Raiber L, Nazareth M, et al. Estrogen disrupts chemokine-mediated chemokine release from mammary cells: Implications for the interplay between estrogen and ip-10 in the regulation of mammary tumor formation. Breast Cancer Res Tr 2004;84:235-45. doi: DOI 10.1023/B:BREA.0000019961.59306.f6
[11] Sutoyo DA, Atmaka DR, Sidabutar L. Dietary factors affecting firmicutes and bacteroidetes ratio in solving obesity problem: A literature review. Media Gizi Indones 2020;15:94-109.
[12] Magne F, Gotteland M, Gauthier L, Zazueta A, Pesoa S, Navarrete P, Balamurugan R. The firmicutes/bacteroidetes ratio: A relevant marker of gut dysbiosis in obese patients? Nutrients 2020;12. doi: 10.3390/nu12051474
[13] Priego T, Sanchez J, Pico C, Palou A. Sex-differential expression of metabolism-related genes in response to a high-fat diet. Obesity (Silver Spring) 2008;16:819-26. doi: 10.1038/oby.2007.117
[14] Eckel LA, Moore SR. Diet-induced hyperphagia in the rat is influenced by sex and exercise. Am J Physiol-Reg I 2004;287:R1080-R5. doi: 10.1152/ajpregu.00424.2004
[15] Taraschenko OD, Maisonneuve IM, Glick SD. Resistance of male sprague–dawley rats to sucrose-induced obesity: Effects of 18-methoxycoronaridine. Physiol Behav 2011;102:126-31.
[16] Roca P, Rodriguez AM, Oliver P, Bonet ML, Quevedo S, Picó C, Palou A. Brown adipose tissue response to cafeteria diet-feeding involves induction of the ucp2 gene and is impaired in female rats as compared to males. Pflügers Archiv 1999;438:628-34.
[17] Stubbins RE, Holcomb VB, Hong J, Núñez NP. Estrogen modulates abdominal adiposity and protects female mice from obesity and impaired glucose tolerance. European journal of nutrition 2012;51:861-70.
[18] Salinero AE, Anderson BM, Zuloaga KL. Sex differences in the metabolic effects of diet-induced obesity vary by age of onset. International journal of obesity 2018;42:1088-91.
[19] Yang Y, Smith Jr DL, Keating KD, Allison DB, Nagy TR. Variations in body weight, food intake and body composition after long‐term high‐fat diet feeding in c57bl/6j mice. Obesity 2014;22:2147-55.
[20] Asarian L, Geary N. Sex differences in the physiology of eating. American Journal of Physiology-Regulatory, Integrative and Comparative Physiology 2013;305:R1215-R67.
[21] Marco A, Schroeder M, Weller A. Microstructural pattern of palatable food intake from weaning to adulthood in male and female oletf rats. Behavioral neuroscience 2009;123:1251.
[22] Strohmayer A, Smith G. The meal pattern of genetically obese (ob/ob) mice. Appetite 1987;8:111-23.
[23] Kim JS, Kirkland RA, Lee SH, Cawthon CR, Rzepka KW, Minaya DM, et al. Gut microbiota composition modulates inflammation and structure of the vagal afferent pathway. Physiol Behav 2020;225. doi: ARTN 113082
10.1016/j.physbeh.2020.113082
[24] Minaya DM, Robertson KL, Rowland NE. Circadian and economic factors affect food acquisition in rats restricted to discrete feeding opportunities. Physiol Behav 2017;181:10-5. doi: 10.1016/j.physbeh.2017.09.003
[25] de Lartigue G, Barbier de la Serre C, Espero E, Lee J, Raybould HE. Leptin resistance in vagal afferent neurons inhibits cholecystokinin signaling and satiation in diet induced obese rats. Plos One 2012;7:e32967. doi: 10.1371/journal.pone.0032967
[26] Tabata Y, Miyao M, Inamoto T, Ishii T, Hirano Y, Yamaoki Y, Ikada Y. De novo formation of adipose tissue by controlled release of basic fibroblast growth factor. Tissue Eng 2000;6:279-89. doi: 10.1089/10763270050044452
Reviewer 4 Report
Comments and Suggestions for Authors
Title: Differential Effects of High-Energy Diet Consumption on Body Composition, Gut Microbiota, and Inflammatory Markers in Male and Female Rats.
This manuscript presents a well-structured study on sex-specific responses to a high-energy diet in Sprague-Dawley rats, an important topic in metabolic and obesity research. The study is methodologically relevant, with clear hypotheses, well-described experimental procedures, and appropriate statistical analyses; figures and tables are well-integrated into the discussion, supporting the key findings; and the discussion provides a thoughtful interpretation of results in the context of existing literature.
However, while the text is generally well-written, it contains awkward phrasing, redundancy, and minor grammatical errors that affect readability. Some claims (e.g. microbiome changes and their functional consequences) could benefit from additional discussion of limitations, certain sections (e.g. Discussion) could be streamlined to improve logical flow, and the manuscript contains inconsistencies in referencing style, spacing, and typographical errors.
On the other hand, the Discussion section could be more concise by reducing redundant explanations.
Mistakes and suggestions by lines:
Title: "Differential Effects of High-Energy Diet Consumption on Body Composition, Gut Microbiota, and Inflammatory Markers in Male and Female Rats." Better: "Sex-Specific Effects of a High-Energy Diet on Body Composition, Gut Microbiota, and Inflammatory Markers in Rats."
Line 13. "Consumption of a high energy density diet (HED) leads to increased body weight and fat mass, gut dysbiosis, and obesity." Better: "A high-energy density diet (HED) promotes body weight gain, fat accumulation, and gut dysbiosis, contributing to obesity."
Lines 21-23. "In addition, male rats displayed either a diet-induced obesity (DIO-P) phenotype or a diet-resistant (DR) phenotype." Please, clarify how these phenotypes were classified.
Line 29. "These findings underscore the importance of considering sex as a biological variable in studies of diet-induced obesity and its metabolic consequences." Better: "Our findings highlight the critical role of sex as a biological variable in diet-induced obesity research."
Line 37. "Obesity is considered a chronic, multifactorial, low-grade inflammatory disease characterized by an excessive increase in fat mass accumulation that can impair health." Better: "Obesity is a chronic, multifactorial disease characterized by excessive fat accumulation and low-grade inflammation, leading to health impairments."
Lines 52-54. "In addition, in a study comparing the gut microbiome of male and female rats exposed to a HED for three weeks, males had a greater increase in the Firmicutes to Bacteroidetes ratio than female rats." Please, specify the reference and clarify the significance of this ratio.
Lines 74-78. "Studies comparing diet-induced obesity between male and female rats and mice are very few and often yield conflicting results." Please, provide examples of conflicting studies.
Line 92. "Male and female Sprague–Dawley rats (n = 6 per group; ~300 g; Envigo, Indianapolis, IN) were housed individually in conventional polycarbonate shoe-box cages..." Please, indicate how sample size was determined (power analysis?).
Line 110. "Blood samples were collected on the last day of LED and last day of HED." Please, specify the method of blood collection (e.g., cardiac puncture, tail vein).
Line 132. "Fecal samples were collected after four weeks of HED consumption." Please, mention storage conditions (e.g., -80 °C).
Line 151. "Consumption of a HED diet significantly increased body weight and fat mass accumulation." Better: "HED consumption significantly increased body weight and fat mass."
Line 186-189. "HED consumption significantly reduced the number of different bacteria species present in the gut microbiota (Two-way ANOVA, F(2,41)=43.21, p<0.0001) in males and females within one week." Please, clarify whether diversity indices (e.g., Shannon, Simpson) were used.
Line 244. "We performed linear regression analysis between identified taxa and the phenotypic outcomes assessed in this study." Please, explain how multiple comparisons were controlled (e.g., Bonferroni correction?).
Line 377. "In the current study, we sought to characterize the initial responses of body weight, body composition, inflammatory markers, gut microbiome, and fat excretion in feces to HED consumption in male and female rats." Better: "This study investigated sex-specific differences in body weight, body composition, inflammatory markers, gut microbiome, and fecal fat excretion following short-term HED consumption in rats."
Line 391. "Fat excretion in feces was significantly higher after HED, and this response was more pronounced in males than females." Please, mention possible mechanisms (e.g., lipid metabolism differences).
Line 534-536. "Differences in inflammatory responses further highlight sex-specific adaptations, with males showing elevated TGF-β levels, while females exhibited increases in IP-10, RANTES, and FGFb." Please, discuss potential physiological implications of these differences.
Lines 543-544. "These findings underscore the importance of considering sex as a biological variable in studies of diet-induced obesity and its metabolic consequences." Better: "Our findings emphasize the need to consider sex differences in obesity research and dietary interventions."
Comments on the Quality of English LanguagePlease, see report
Author Response
We thank the Reviewer for the comments, we have addressed them to the best of our abilities and believe the manuscript have been significantly improved. Our response to each specific comment is presented below (in blue). In addition, revisions to the manuscript have been highlighted in yellow so editors and reviewers can readily see changes made.
This manuscript presents a well-structured study on sex-specific responses to a high-energy diet in Sprague-Dawley rats, an important topic in metabolic and obesity research. The study is methodologically relevant, with clear hypotheses, well-described experimental procedures, and appropriate statistical analyses; figures and tables are well-integrated into the discussion, supporting the key findings; and the discussion provides a thoughtful interpretation of results in the context of existing literature.
However, while the text is generally well-written, it contains awkward phrasing, redundancy, and minor grammatical errors that affect readability. Some claims (e.g. microbiome changes and their functional consequences) could benefit from additional discussion of limitations, certain sections (e.g. Discussion) could be streamlined to improve logical flow, and the manuscript contains inconsistencies in referencing style, spacing, and typographical errors.
On the other hand, the Discussion section could be more concise by reducing redundant explanations.
Response:
We appreciate the attention to detail. The manuscript has been edited to improve clarity and readability. In addition, the journal’s endnote file has been used to ensure consistency in the referencing style.
Mistakes and suggestions by lines:
Title: "Differential Effects of High-Energy Diet Consumption on Body Composition, Gut Microbiota, and Inflammatory Markers in Male and Female Rats." Better: "Sex-Specific Effects of a High-Energy Diet on Body Composition, Gut Microbiota, and Inflammatory Markers in Rats."
Response:
We appreciate the suggestion and the title has been modified as recommended.
Line 13. "Consumption of a high energy density diet (HED) leads to increased body weight and fat mass, gut dysbiosis, and obesity." Better: "A high-energy density diet (HED) promotes body weight gain, fat accumulation, and gut dysbiosis, contributing to obesity."
Response:
We appreciate the suggestion. The text has been modified as recommended.
Lines 21-23. "In addition, male rats displayed either a diet-induced obesity (DIO-P) phenotype or a diet-resistant (DR) phenotype." Please, clarify how these phenotypes were classified.
Response:
The sentence has been modified to read: “Males displayed either a diet-induced obesity (DIO-P) or a diet-resistant (DR) phenotype, as characterized by their differential body weight gain.”
Line 29. "These findings underscore the importance of considering sex as a biological variable in studies of diet-induced obesity and its metabolic consequences." Better: "Our findings highlight the critical role of sex as a biological variable in diet-induced obesity research."
Response:
We appreciate the suggestion. The text has been modified as recommended.
Line 37. "Obesity is considered a chronic, multifactorial, low-grade inflammatory disease characterized by an excessive increase in fat mass accumulation that can impair health." Better: "Obesity is a chronic, multifactorial disease characterized by excessive fat accumulation and low-grade inflammation, leading to health impairments."
Response:
We appreciate the suggestion. The text has been modified as recommended.
Lines 52-54. "In addition, in a study comparing the gut microbiome of male and female rats exposed to a HED for three weeks, males had a greater increase in the Firmicutes to Bacteroidetes ratio than female rats." Please, specify the reference and clarify the significance of this ratio.
Response:
The following text has been added to clarify the significance of this ration:
“The Firmicutes/Bacteroidetes ration has been proposed as a potential biomarker for obesity because studies have shown that the gut microbiota composition of obese animals and humans exhibits an increased in abundance of firmicutes, decreased abundance of Bacteroidetes compared to lean counterparts [11, 12]”
Lines 74-78. "Studies comparing diet-induced obesity between male and female rats and mice are very few and often yield conflicting results." Please, provide examples of conflicting studies.
Response:
The following text has been added to the introduction section:
“It has been reported that female rats have are more susceptible to metabolic challenges [13-16], while research in mice generally indicates that females are either equally or less prone to dietary obesity [17-19]. Male rats and mice consume more food than their female counterparts, but the mechanisms or eating patterns underlying this sex difference may vary between species. Male rats typically consume larger meals, whereas male mice eat meals of the same size as female mice, but more frequently [20-22].”
Line 92. "Male and female Sprague–Dawley rats (n = 6 per group; ~300 g; Envigo, Indianapolis, IN) were housed individually in conventional polycarbonate shoe-box cages..." Please, indicate how sample size was determined (power analysis?).
Response:
We acknowledge that a formal power analysis was not conducted for this study. Sample size was determined based on prior studies from our laboratory and taking into account similar studies that reported comparable sample sizes [2, 23-25].
Line 110. "Blood samples were collected on the last day of LED and last day of HED." Please, specify the method of blood collection (e.g., cardiac puncture, tail vein).
Response:
The sentence has been modified to read: “Blood samples were collected on the last day of LED and last day of HED by cardiac puncture.”
Line 132. "Fecal samples were collected after four weeks of HED consumption." Please, mention storage conditions (e.g., -80 °C).
Response:
The sentence has been modified to read: “Fecal samples were collected after four weeks of HED consumption and stored at −21°C.”
Line 151. "Consumption of a HED diet significantly increased body weight and fat mass accumulation." Better: "HED consumption significantly increased body weight and fat mass."
Response:
We appreciate the suggestion. The text has been modified as recommended.
Line 186-189. "HED consumption significantly reduced the number of different bacteria species present in the gut microbiota (Two-way ANOVA, F(2,41)=43.21, p<0.0001) in males and females within one week." Please, clarify whether diversity indices (e.g., Shannon, Simpson) were used.
Response:
The sentence has been modified to read: “HED consumption significantly reduced the number of different bacteria species present in the gut microbiota as evidenced by the Shannon index (Two-way ANOVA, F(2,41)=43.21, p<0.0001) in males and females within one week”
Line 244. "We performed linear regression analysis between identified taxa and the phenotypic outcomes assessed in this study." Please, explain how multiple comparisons were controlled (e.g., Bonferroni correction?).
Response:
We used simple regression analysis examining the relationship between two variables only -abundance for a given bacterial taxa identified, e.g. caccae and a phenotypic outcome measured, e.g. body weight. Thus, it was not needed to control for multiple comparisons.
The sentence has been modified to read: “We performed simple linear regression analysis between identified taxa and the phenotypic outcomes assessed in this study”
Line 377. "In the current study, we sought to characterize the initial responses of body weight, body composition, inflammatory markers, gut microbiome, and fat excretion in feces to HED consumption in male and female rats." Better: "This study investigated sex-specific differences in body weight, body composition, inflammatory markers, gut microbiome, and fecal fat excretion following short-term HED consumption in rats."
Response:
We appreciate the suggestion. The text has been modified as recommended.
Line 391. "Fat excretion in feces was significantly higher after HED, and this response was more pronounced in males than females." Please, mention possible mechanisms (e.g., lipid metabolism differences).
Response:
The following text has been added at the end of that sentence:
“This discrepancy may be explained by sex differences in lipid absorption and metabolism.” In addition, specific details on lipid metabolism are further discussed in lines 482-492.
Line 534-536. "Differences in inflammatory responses further highlight sex-specific adaptations, with males showing elevated TGF-β levels, while females exhibited increases in IP-10, RANTES, and FGFb." Please, discuss potential physiological implications of these differences.
Response:
We appreciate the recommendation. The discussion section has been modified as follows:
TGF-β: The following text has been added to the discussion section:
“It is possible that the increase level of TGF-β observed in males only is mediated by testosterone as previous studies have shown that testosterone can increase plasma levels of TGF-β [5-7].”
RANTES: We observed a significant decrease in RANTES levels in females following the HED diet. The following text has been added to the discussion section:
“However, a previous study in female rats exposed to a high cholesterol diet revealed that estrogen suppressed diet-induced RANTES expression in hippocampal tissue [8].”
IP-10: The following text has been added to the discussion section:
“The observed difference in response between males and females may be partially mediated by estrogen. IP-10 induces the release of additional proinflammatory cytokines, and this response is attenuated in the presence of estrogen [9]. It is plausible that the elevated levels of IP-10 serve as a compensatory mechanism to ensure the proper magnitude of the inflammatory response.”
FGFb: The following text has been added to the discussion section:
“HED had no significant impact on FGFb levels in rats. Under LED conditions, male rats exhibited slightly lower levels than females. This difference became more pronounced on HED largely due to a slight decrease in FGFb levels in DR male rats. There is limited research on the involvement of FGFb in obesity development and progression in rodent models and human subjects. However, FGFb is known to induce strong vascularization and has been reported to induce the generation of new fat cells in mice [26]. It is possible that the effects of HED consumption on FGFb are tissue specific rather than systemic, suggesting the need for further research.”
Lines 543-544. "These findings underscore the importance of considering sex as a biological variable in studies of diet-induced obesity and its metabolic consequences." Better: "Our findings emphasize the need to consider sex differences in obesity research and dietary interventions."
Response:
We appreciate the suggestion. The text has been modified as recommended.
References
[1] Levin BE, Dunn-Meynell AA, Balkan B, Keesey RE. Selective breeding for diet-induced obesity and resistance in sprague-dawley rats. Am J Physiol 1997;273:R725-30. doi: 10.1152/ajpregu.1997.273.2.R725
[2] Minaya DM, Turlej A, Joshi A, Nagy T, Weinstein N, DiLorenzo P, et al. Consumption of a high energy density diet triggers microbiota dysbiosis, hepatic lipidosis, and microglia activation in the nucleus of the solitary tract in rats. Nutr Diabetes 2020;10. doi: ARTN 20
10.1038/s41387-020-0119-4
[3] Shin NR, Lee JC, Lee HY, Kim MS, Whon TW, Lee MS, Bae JW. An increase in the akkermansia spp. Population induced by metformin treatment improves glucose homeostasis in diet-induced obese mice. Gut 2014;63:727-35. doi: 10.1136/gutjnl-2012-303839
[4] Everard A, Belzer C, Geurts L, Ouwerkerk JP, Druart C, Bindels LB, et al. Cross-talk between akkermansia muciniphila and intestinal epithelium controls diet-induced obesity. Proceedings of the national academy of sciences 2013;110:9066-71.
[5] Zhao S, Liu W, Wang J, Shi J, Sun Y, Wang W, et al. Akkermansia muciniphila improves metabolic profiles by reducing inflammation in chow diet-fed mice. J Mol Endocrinol 2017;58:1-14. doi: 10.1530/JME-16-0054
[6] Mendes LO, Scarano WR, Rochel-Maia SS, Fioruci-Fontaneli BA, Chuffa LG, Martinez FE. Testosterone therapy differently regulates the anti- and pro-inflammatory cytokines in the plasma and prostate of rats submitted to chronic ethanol consumption (uchb). Am J Reprod Immunol 2014;72:317-25. doi: 10.1111/aji.12268
[7] Liva SM, Voskuhl RR. Testosterone acts directly on cd4+ t lymphocytes to increase il-10 production. J Immunol 2001;167:2060-7. doi: 10.4049/jimmunol.167.4.2060
[8] De Paula GC, Simões RF, Garcia-Serrano AM, Duarte JM. High-fat and high-sucrose diet-induced hypothalamic inflammation shows sex specific features in mice. Neurochemical Research 2024;49:3356-66.
[9] Lewis DK, Bake S, Thomas K, Jezierski MK, Sohrabji F. A high cholesterol diet elevates hippocampal cytokine expression in an age and estrogen-dependent manner in female rats. J Neuroimmunol 2010;223:31-8. doi: 10.1016/j.jneuroim.2010.03.024
[10] Aronica SM, Fanti P, Kaminskaya K, Gibbs K, Raiber L, Nazareth M, et al. Estrogen disrupts chemokine-mediated chemokine release from mammary cells: Implications for the interplay between estrogen and ip-10 in the regulation of mammary tumor formation. Breast Cancer Res Tr 2004;84:235-45. doi: DOI 10.1023/B:BREA.0000019961.59306.f6
[11] Sutoyo DA, Atmaka DR, Sidabutar L. Dietary factors affecting firmicutes and bacteroidetes ratio in solving obesity problem: A literature review. Media Gizi Indones 2020;15:94-109.
[12] Magne F, Gotteland M, Gauthier L, Zazueta A, Pesoa S, Navarrete P, Balamurugan R. The firmicutes/bacteroidetes ratio: A relevant marker of gut dysbiosis in obese patients? Nutrients 2020;12. doi: 10.3390/nu12051474
[13] Priego T, Sanchez J, Pico C, Palou A. Sex-differential expression of metabolism-related genes in response to a high-fat diet. Obesity (Silver Spring) 2008;16:819-26. doi: 10.1038/oby.2007.117
[14] Eckel LA, Moore SR. Diet-induced hyperphagia in the rat is influenced by sex and exercise. Am J Physiol-Reg I 2004;287:R1080-R5. doi: 10.1152/ajpregu.00424.2004
[15] Taraschenko OD, Maisonneuve IM, Glick SD. Resistance of male sprague–dawley rats to sucrose-induced obesity: Effects of 18-methoxycoronaridine. Physiol Behav 2011;102:126-31.
[16] Roca P, Rodriguez AM, Oliver P, Bonet ML, Quevedo S, Picó C, Palou A. Brown adipose tissue response to cafeteria diet-feeding involves induction of the ucp2 gene and is impaired in female rats as compared to males. Pflügers Archiv 1999;438:628-34.
[17] Stubbins RE, Holcomb VB, Hong J, Núñez NP. Estrogen modulates abdominal adiposity and protects female mice from obesity and impaired glucose tolerance. European journal of nutrition 2012;51:861-70.
[18] Salinero AE, Anderson BM, Zuloaga KL. Sex differences in the metabolic effects of diet-induced obesity vary by age of onset. International journal of obesity 2018;42:1088-91.
[19] Yang Y, Smith Jr DL, Keating KD, Allison DB, Nagy TR. Variations in body weight, food intake and body composition after long‐term high‐fat diet feeding in c57bl/6j mice. Obesity 2014;22:2147-55.
[20] Asarian L, Geary N. Sex differences in the physiology of eating. American Journal of Physiology-Regulatory, Integrative and Comparative Physiology 2013;305:R1215-R67.
[21] Marco A, Schroeder M, Weller A. Microstructural pattern of palatable food intake from weaning to adulthood in male and female oletf rats. Behavioral neuroscience 2009;123:1251.
[22] Strohmayer A, Smith G. The meal pattern of genetically obese (ob/ob) mice. Appetite 1987;8:111-23.
[23] Kim JS, Kirkland RA, Lee SH, Cawthon CR, Rzepka KW, Minaya DM, et al. Gut microbiota composition modulates inflammation and structure of the vagal afferent pathway. Physiol Behav 2020;225. doi: ARTN 113082
10.1016/j.physbeh.2020.113082
[24] Minaya DM, Robertson KL, Rowland NE. Circadian and economic factors affect food acquisition in rats restricted to discrete feeding opportunities. Physiol Behav 2017;181:10-5. doi: 10.1016/j.physbeh.2017.09.003
[25] de Lartigue G, Barbier de la Serre C, Espero E, Lee J, Raybould HE. Leptin resistance in vagal afferent neurons inhibits cholecystokinin signaling and satiation in diet induced obese rats. Plos One 2012;7:e32967. doi: 10.1371/journal.pone.0032967
[26] Tabata Y, Miyao M, Inamoto T, Ishii T, Hirano Y, Yamaoki Y, Ikada Y. De novo formation of adipose tissue by controlled release of basic fibroblast growth factor. Tissue Eng 2000;6:279-89. doi: 10.1089/10763270050044452
Round 2
Reviewer 1 Report
Comments and Suggestions for Authors
This reviewer still has questions.
- Is it common to divide dio-p and dr into two groups, one with the highest weight gain and one with the lowest weight gain?
- Not adding a control group that continues to eat LED should at least be mentioned in the limitation.
- The dio-p and dio-d gut microbiota should not be grouped together, if there is no difference in their gut microbiota.
Author Response
We thank the reviewers for their comments, we have addressed them to the best of our abilities and believe the manuscript have been significantly improved. Our response to each specific comment is presented below. In addition, revisions to the manuscript have been highlighted in yellow so editors and reviewers can readily see changes made.
Round 2
Reviewer 1
Is it common to divide dio-p and dr into two groups, one with the highest weight gain and one with the lowest weight gain?
Response:
Early studies, as well as more recent studies, describing the phenotypes of diet-induced obesity susceptibility and resistance in rats were based on body weight gain observed after exposing animals to a high-calorie diet for a period of typically four weeks or more [1-5].
Not adding a control group that continues to eat LED should at least be mentioned in the limitation.
Response:
We appreciate the recommendation. This limitation is mentioned in the conclusion section as follows: “First, the absence of a control diet (LED) cohort for the entire duration of the study limits the ability to compare long-term effects directly.”
The dio-p and dio-d gut microbiota should not be grouped together, if there is no difference in their gut microbiota.
Response:
We appreciate the suggestion. Microbiota data is now presented separately for DR, DIO-P, and female rats (Figure 2). The corresponding results and discussion sections have been updated accordingly.
References:
[1] Chang S, Graham B, Yakubu F, Lin D, Peters JC, Hill JO. Metabolic differences between obesity-prone and obesity-resistant rats. Am J Physiol 1990;259:R1103-R10. doi: DOI 10.1152/ajpregu.1990.259.6.R1103
[2] Levin BE, Triscari J, Hogan S, Sullivan AC. Resistance to diet-induced obesity - food-intake, pancreatic sympathetic tone, and insulin. Am J Physiol 1987;252:R471-R8. doi: DOI 10.1152/ajpregu.1987.252.3.R471
[3] Tschöp M, Heiman ML. Rodent obesity models:: An overview. Exp Clin Endocr Diab 2001;109:307-19. doi: DOI 10.1055/s-2001-17297
[4] de Lartigue G, de la Serre CB, Espero E, Lee J, Raybould HE. Diet-induced obesity leads to the development of leptin resistance in vagal afferent neurons. Am J Physiol-Endoc M 2011;301:E187-E95. doi: 10.1152/ajpendo.00056.2011
[5] Paulino G, de la Serre CB, Knotts TA, Oort PJ, Newman JW, Adams SH, Raybould HE. Increased expression of receptors for orexigenic factors in nodose ganglion of diet-induced obese rats. Am J Physiol-Endoc M 2009;296:E898-E903. doi: 10.1152/ajpendo.90796.2008
Reviewer 3 Report
Comments and Suggestions for Authors
The authors have fully addressed all of my comments. However, I feel that the manuscript requires minor stylistic corrections. In particular, the figure titles do not seem to follow the journal's guidelines.
Author Response
We thank the reviewers for their comments, we have addressed them to the best of our abilities and believe the manuscript have been significantly improved. Our response to each specific comment is presented below (in orange). In addition, revisions to the manuscript have been highlighted in yellow so editors and reviewers can readily see changes made.
Reviewer 3
The authors have fully addressed all of my comments. However, I feel that the manuscript requires minor stylistic corrections. In particular, the figure titles do not seem to follow the journal's guidelines.
Response:
We appreciate the attention to details. Figure titles have been revised to comply with journal’s guidelines.
Round 3
Reviewer 1 Report
Comments and Suggestions for Authors
No further comments.